# 1 Modeling impacts of ozone on gross primary production across

## **European forest ecosystems using JULES**

- 3 Inês Vieira<sup>1\*</sup>, Félicien Meunier<sup>1,2</sup>, Maria Carolina Duran Rojas<sup>3</sup>, Stephen Sitch<sup>3</sup>, Flossie Brown<sup>4</sup>,
- 4 Giacomo Gerosa<sup>5</sup>, Silvano Fares<sup>6</sup>, Pascal Boeckx<sup>2</sup>, Marijn Bauters<sup>1</sup> and Hans Verbeeck<sup>1</sup>
- <sup>1</sup> Q-ForestLab, Laboratory of Quantitative Forest Ecosystem Science, Department of Environment, Ghent University, Gent,
- 7 Belgium

5

- <sup>2</sup>ISOFYS, Isotope Bioscience Laboratory, Department of Green Chemistry and Technology, Ghent University, Gent, Belgium
- <sup>3</sup> Faculty of Environment, Science & Economy, University of Exeter, Exeter, United Kingdom
- <sup>4</sup>Institute for Atmospheric and Climate Science, ETH Zurich, Zurich, Switzerland
  - <sup>5</sup> Faculty of Mathematical, Physical and Natural Sciences, Università Cattolica del Sacro Cuore, Brescia, Italy
- 6 Institute for Agriculture and Forestry Systems in the Mediterranean, National Research Council of Italy, Naples, Italy
- Correspondence to: Inês Vieira (ines.dossantosvieira@ugent.be)
- Abstract. This study investigates the effects of tropospheric ozone (O<sub>3</sub>), a potent greenhouse gas and air pollutant, on European forests, an issue lacking comprehensive analysis at the site level. Unlike other greenhouse gases, tropospheric O<sub>3</sub> is primarily
- formed through photochemical reactions, and it significantly impairs vegetation productivity and carbon fixation, thereby
- affecting forest health and ecosystem services. We utilise data from multiple European flux tower sites and integrate statistical
- and mechanistic modelling approaches to simulate O<sub>3</sub> impacts on photosynthesis and stomatal conductance. The study
- examines six key forest sites across Europe: Hyvtiälä and Värriö (Finland), Brasschaat (Belgium), Fontainebleau-Barbeau
- (France), Bosco-Fontana, and Castelporziano 2 (Italy), representing boreal, temperate, and Mediterranean climates. These sites
- provide a diverse range of environmental conditions and forest types, enabling a comprehensive assessment of O<sub>3</sub> effects on
- Gross Primary Production (GPP). We calibrated the Joint UK Land Environment Simulator (JULES) model using observed
- GPP data to simulate different O<sub>3</sub> exposure sensitivities. Incorporating O<sub>3</sub> effects improved the model's accuracy across all
- sites, although the magnitude of improvement varied depending on site-specific factors such as vegetation type, climate, and
- ozone exposure levels. The GPP reduction due to ozone exposure varied considerably across sites, with annual mean reductions
- ranging from 1.04% at Värriö to 6.2% at Bosco-Fontana. These findings emphasise the need to account for local environmental
- conditions when assessing ozone stress on forests. This study highlights key model strengths and limitations in representing
- O<sub>3</sub>-vegetation interactions, with implications for improving forest productivity simulations under future air pollution
- scenarios. The model effectively captures the diurnal and seasonal variability of GPP and its sensitivity to O<sub>3</sub> stress, particularly

in boreal and temperate forests. However, its performance is limited in Mediterranean ecosystems, where pronounced O<sub>3</sub> peaks and environmental stressors such as high vapor pressure deficit exacerbate GPP declines, pointing to the need for improved parameterisation and representation of site-specific processes. By integrating in situ measurements, this research contributes to developing targeted strategies for mitigating the adverse effects of O<sub>3</sub> on forest ecosystems.

#### 1 Introduction

Ground-level ozone (O3) is a greenhouse gas and an air pollutant with a strong oxidative capacity being responsible for 35 36 negatively impacting human health (Nuvolone et al., 2018; Lu and Yao, 2023), water and carbon cycles (Sitch et al., 2007; 37 Lombardozzi et al., 2015), agriculture and crop production (Van Dingenen et al., 2009; Feng et al., 2022; Li et al., 2022) and 38 vegetation productivity (Ainsworth et al., 2012; Yue and Unger, 2014; Ainsworth et al., 2019; Savi et al., 2020). In the 39 troposphere, O<sub>3</sub> is not emitted directly, contrary to other greenhouse gases, such as carbon dioxide (CO<sub>2</sub>) and methane (CH<sub>4</sub>). 40 The majority of O<sub>3</sub> (about 90%) is generated by the photochemical oxidation of its precursor gases (natural and anthropogenic), 41 such as CH<sub>4</sub>, carbon monoxide (CO), and volatile carbon compounds (VOCs) in the presence of nitrogen oxides (NOx). The 42 remaining 10% is from the influx of ozone from the stratosphere. On the other hand, tropospheric O<sub>3</sub> is primarily removed 43 through chemical destruction and dry deposition to terrestrial surfaces that occurs via stomatal (Fowler et al., 2009; Ducker et 44 al., 2018; Clifton et al., 2020) and non-stomatal pathways (Zhang et al., 2003; Wong et al., 2022). 45 Stomatal O<sub>3</sub> uptake damages vegetation by causing cell death and decreasing carbon fixation (Li et al., 2019), which in turn 46 leads to reduced productivity (Ainsworth et al., 2012) and early senescence (Gielen et al., 2007). In particular, it reduces gross 47 primary production (GPP), the gross carbon uptake via photosynthesis, a measure of ecosystem productivity (Proietti et al., 2016; Cailleret et al., 2018; Grulke et al., 2019). Exposure to O<sub>3</sub> leads to reductions in photosynthesis and stomatal 48 49 conductance, thereby decreasing both gross primary productivity (GPP) and transpiration. These physiological impacts have 50 broader consequences for climate, including reduced carbon uptake, decreased latent heat flux (LE), and diminished water 51 vapour release. Additionally, lower stomatal conductance reduces dry deposition of ozone, which can exacerbate near-surface 52 ozone concentrations. Therefore, incorporating a representation of ozone damage to plants in land surface and Earth System 53 models (LSMs and ESMs) is essential because many regions experience potentially damaging O<sub>3</sub> concentrations. However, 54 while most studies agree that O<sub>3</sub> exposure results in significant reductions in GPP, the reduction varies with measurement 55 location or assumptions used in the models. For example, Sitch et al. (2007) predicted a decline in global GPP of 14 to 23% 56 by 2100. Lombardozzi et al. (2015) predicted a 10.8% decrease in present-day (2002–2009) GPP globally. Similarly, Yue and 57 Unger (2014) reported that ozone damage reduced GPP by an average of 4–8% across the eastern United States, with localised 58 reductions reaching as high as 11–17% along the east coast. 59 In Europe, surface O<sub>3</sub> pollution poses a significant challenge to air quality, particularly in southern Europe, where high solar 60 radiation and anthropogenic emissions—mainly from traffic and industrial activity—enhance photochemical O<sub>3</sub> formation

(Sicard et al., 2021). Currently, the European standard used to protect vegetation against negative impacts of O<sub>3</sub> is the

Accumulated Ozone over a Threshold of 40 ppb (AOT40), i.e. the cumulative exposure to hourly O<sub>3</sub> concentrations above 40 ppb over the daylight hours of the growing season (Anay et al., 2017; Projetti et al., 2021). However, the O<sub>3</sub> uptake through stomata is a better metric for assessing plant damage because it estimates the actual quantity of O<sub>3</sub> entering the leaf tissues (Anav et al., 2016; Sicard et al., 2016). High ambient O<sub>3</sub> levels may not damage plants when drought and/or other environmental stressors limit the stomatal aperture (Shang et al., 2024). Therefore, flux-based approaches were developed to assess the effects of O<sub>3</sub> on vegetation. This method quantifies leaf O<sub>3</sub> uptake and the dose that actually enters the plant tissue via stomata and considers the environmental constraints that may limit optimal stomatal conductance. For example, Projetti et al. (2016) performed a comprehensive study on 37 European forest sites during the period of 2000-2010 to assess surface O<sub>3</sub> effects on GPP. In this study, the DO<sub>3</sub>SE (Deposition of O<sub>3</sub> and Stomatal Exchange) model (Emberson et al., 2001) was used to estimate ozone uptake/stomatal ozone flux using the Jarvis multiplicative method for stomatal conductance (Jarvis, 1976). The results showed that GPP was reduced between 0.4% and 30% annually across different sites. Also, Anav et al. (2011) showed, using a land surface model coupled with a chemistry transport model, a 22% reduction in yearly GPP and a 15-20% reduction in leaf area index (LAI) due to O<sub>3</sub> exposure, with the most substantial impacts occurring during the summer months. Interestingly, not all studies have found significant negative effects of O<sub>3</sub> on GPP. For instance, research on a Scots pine stand in Belgium over 15 years found no significant O<sub>3</sub> effects on GPP despite high stomatal O<sub>3</sub> uptake (Verryckt et al., 2017). This suggests that the impact of O<sub>3</sub> may vary depending on specific forest types (Sorrentino et al., 2025) and local conditions (Lin et al., 2019; Otu-Larbi et al. 2020). Satellite observations have also been utilised to assess O<sub>3</sub>-induced GPP reductions. estimating a decrease of 0.4-9.6% across European forests from 2003-2015. These findings align with previous estimates and highlight soil moisture as a critical interacting variable influencing GPP reductions, particularly in Mediterranean regions (Vargas et al., 2013). Therefore, while the negative effects of O<sub>3</sub> on GPP in European forests are well-documented, the extent of these impacts can vary significantly based on regional conditions, forest types, methodological approaches, and it is not clear what drives the local differences. Understanding these variations is crucial for accurately assessing the broader implications of O<sub>3</sub> on forest productivity and ecosystem services. This gap in the literature underscores the need for detailed studies that evaluate the influence of ozone on forest productivity in Europe using advanced process-based models. This study provides a detailed, site-level analysis of O<sub>3</sub> impacts on GPP across European forests, leveraging local in situ measurements of O<sub>3</sub>, CO<sub>2</sub> exchange, and meteorological data to optimise the Joint UK Land Environment Simulator (JULES) model. Our objectives are to quantify O<sub>3</sub>-induced GPP limitations and assess model improvements through the incorporation of ozone damage mechanisms. Specifically, we aim to address the following research questions:

- 1. To what extent can we improve GPP simulations for European forests of a process-based model by incorporating plant sensitivity to ozone?
- 2. To what extent does ozone limit GPP across European forests?

65

67

70

72

3. How do ozone impacts interact with other environmental factors, and how can an optimised model help us understand these mechanisms, particularly on high-ozone days?

To achieve these objectives, we combined a multi-year eddy covariance flux tower dataset across a latitudinal gradient in

Europe across six sites in boreal, temperate, and Mediterranean forests and statistical and process-based models, providing a

comprehensive understanding of ozone's effects on GPP.

#### 2 Materials and Methods

#### 2.1 Study Area

116117

120121122

We investigated six sites along a European latitudinal gradient in four countries: Finland, Belgium, France and Italy (Fig. 1,

Table 1), belonging to the Integrated Carbon Observation System (ICOS, https://www.icos-cp.eu/, last access: 20 September

2024). These sites span boreal, temperate, and Mediterranean climates, representing diverse forest ecosystems with varying

ozone exposure, productivity, and environmental conditions.

The Värriö site (FI-Var) of the University of Helsinki is located in Värriö strict nature reserve, Salla, Finnish Lapland. The area lies 130 km north of the Arctic Circle and 6 km from the Finnish Russian border. The flux tower is located at the arcticalpine timberline on the top plateau of the hill of Kotovaara, at 395 m a.s.l, and surrounded by a homogeneous and relatively open 10m tall Scots Pine (Pinus sylvestris L.) forest. The leaf area index (LAI) varies between 0.0013 and 0.68 m<sup>2</sup>m<sup>-2</sup> (Dengel et al., 2013). The Hyytiälä forest (FI-Hyy) boreal site is located 220 km NW from Helsinki, Finland. The station is dominated by Scots pine (Pinus sylvestris L.), Norway spruce, and birch on a slightly hilly terrain. The LAI varies between 0.45 and 3.04 m<sup>2</sup>m<sup>-2</sup> (Schraik et al., 2023). The Brasschaat site (BE-Bra) is a forest located 20 km northeast of Antwerp, Belgium. The study site consists of a 150-ha mixed coniferous/deciduous forest dominated by Scots pine. The LAI varies between 1 and 1.5 m<sup>2</sup>m<sup>-</sup> <sup>2</sup> (Op de Beeck et al., 2010). Fontainebleau-Barbeau forest (FR-Fon) is located 53 km southeast of Paris, France. Fontainebleau-Barbeau is a deciduous forest mainly composed of mature sessile oak (Ouercus petraea (Matt.) Liebl). The average LAI over the 2012-2018 period was 5.8 m<sup>2</sup>m<sup>-2</sup>, ranging from 4.6 to 6.8 m<sup>2</sup>m<sup>-2</sup> (Soudani et al., 2021). The Bosco-Fontana site (IT-BFt) is a 233-ha forest composed mainly of mature Oak-Hornbeam (Carpinus betulus) at Po Valley, a few kilometres from Mantova, Italy. The LAI ranges between 0.9 and 3.0 m<sup>2</sup>m<sup>-2</sup> (Gerosa et al., 2022). The Castelporziano 2 site (IT-Cp2) is located in the Presidential Estate of Castelporziano, around 25 km southwest of the centre of Rome, Italy. Castelporziano covers an area of about 6000 ha of undisturbed Mediterranean maquis, oak and pine forests. The experimental site is located inside a pure Holm Oak (Quercus ilex) stand with some shrubs in the understory. The LAI varies between 0.5 and 4.5 m<sup>2</sup>m<sup>-2</sup> (Gratani and Crescente, 2000), More details about each site are available in Table 1.

## Table 1: Overview of the study sites.

| Site | Värriö | Hyytiälä | Brasschaa | Fontainebleau | Bosco-  | Castelporzian |  |
|------|--------|----------|-----------|---------------|---------|---------------|--|
|      |        |          | t         | -Barbeau      | Fontana | o 2           |  |
|      |        |          |           |               |         |               |  |

| Acronym                             | FI-Var                                  | FI-Hyy                                  | BE-Bra                        | FR-Fon                            | IT-BFt                            | IT-Cp2                               |
|-------------------------------------|-----------------------------------------|-----------------------------------------|-------------------------------|-----------------------------------|-----------------------------------|--------------------------------------|
| Country                             | Finland                                 | Finland                                 | Belgium                       | France                            | Italy                             | Italy                                |
| Latitude (°)                        | 67.75                                   | 61.85                                   | 51.30                         | 48.47                             | 45.19                             | 41.70                                |
| Longitude (°)                       | 29.61                                   | 24.29                                   | 4.52                          | 2.78                              | 10.74                             | 12.36                                |
| Elevation (m<br>a.s.l.)             | 395                                     | 181                                     | 16                            | 103                               | 23                                | 19                                   |
| Köppen-<br>Geiger<br>classification | Subarctic<br>(Dfc)                      | Subarctic<br>(Dfc)                      | Marine<br>west coast<br>(Cfb) | Marine west coast (Cfb)           | Humid<br>subtropica<br>l (Cfa)    | Hot-summer<br>Mediterranean<br>(Csa) |
| Forest type                         | Evergree<br>n<br>Needlelea<br>f Forests | Evergree<br>n<br>Needlelea<br>f Forests | Mixed<br>Forests              | Deciduous<br>Broadleaf<br>Forests | Deciduous<br>Broadleaf<br>Forests | Evergreen<br>Broadleaf<br>Forest     |
| Meteorologica<br>l dataset          | 2017-<br>2023                           | 1996-<br>present                        | 1996-<br>present              | 2005-present                      | 2013-<br>2020                     | 2012-present                         |
| 03<br>concentration                 | 2017-<br>2023                           | 1996-<br>present                        | 1996-2020                     | 2014-2020                         | 2013-<br>2020                     | 2013-2014                            |
| Fluxes (GPP,<br>LE)                 | 2017-<br>2023                           | 1996-<br>present                        | 1999-<br>present              | 2005-present                      | 2012-<br>present                  | 2012-present                         |
| Mean annual temperature (°C)        | -0.5                                    | 3.5                                     | 10.5                          | 11.4                              | 14.5                              | 16.43                                |
| Mean annual precipitation (mm)      | 601.0                                   | 711.0                                   | 920.7                         | 678.9                             | 697.0                             | 601.0                                |
| Mean annual 03 concentration (ppb)  | 31.85                                   | 28.37                                   | 23.78                         | 30.08                             | 34.47                             | 27.72                                |

| Maximum 03 concentration (ppb)                                | 109.57                           | 89.32                                | 143.0                                 | 139.25                        | 144.71                          | 119.84                    |
|---------------------------------------------------------------|----------------------------------|--------------------------------------|---------------------------------------|-------------------------------|---------------------------------|---------------------------|
| Mean summer<br>AOT40<br>(ppb.hours)                           | 336.96                           | 1538.42                              | 5406.77                               | 41912.28                      | 20084.90                        | 13172.03                  |
| Mean summer<br>03 (ppb)                                       | 28.37                            | 32.17                                | 31.67                                 | 71.71                         | 42.30                           | 45.98                     |
| Mean annual<br>GPP (tC ha <sup>-1</sup><br>yr <sup>-1</sup> ) | 470.1                            | 470.1                                | 1181                                  | 1452.9                        | 2069.3                          | 1683.6                    |
| Peak LAI<br>(m2m-2)                                           | 0.68<br>(Dengel et<br>al., 2013) | 3.04<br>(Schraik<br>et al.,<br>2023) | 1.31 (Op<br>de Beeck et<br>al., 2010) | 6.8 (Soudani et<br>al., 2021) | 3.0<br>(Gerosa et<br>al., 2022) | 4.76 (Fares et al., 2013) |

Figure 1: Geographical location of the six study sites across Europe: (a) Värriö, Finland (FI-Var),(b) Hyytiälä, Finland (FI-Hyy), (c) Brasschaat, Belgium (BE-Bra), (d) Fontainebleau-Barbeau, France (FR-Fon), (e) Bosco-Fontana, Italy (IT-BFt) and (f) Castelporziano 2, Italy (IT-Cp2). All photos were retrieved from the ICOS website (<a href="https://www.icos-cp.eu/">https://www.icos-cp.eu/</a>, last access: 20 September 2024)

#### 2.2 Meteorological, Ozone, and Ecosystem Flux Datasets

For each site, the following meteorological variables were available on the ICOS data portal: air temperature (TA, °C), relative humidity (RH, %), short-wave radiation (SW, Wm<sup>-2</sup>), precipitation (P, mm), atmospheric pressure (PA, kPa) and vapour pressure deficit (VPD, kPa). Measured half-hourly O<sub>3</sub> concentration data (ppb, Fig. 2) were provided by site principal investigators. The half-hourly Gross Primary Production (GPP, μmol m<sup>-2</sup> s<sup>-1</sup>) and Latent Heat flux (LE, W m<sup>-2</sup>) were derived from eddy covariance measurements at each site. GPP was estimated from net ecosystem exchange (NEE) using standard partitioning techniques implemented in the ICOS ONEFlux processing pipeline (Warm Winter 2020 Team, ICOS Ecosystem Thematic Centre, 2022). LE was derived from water vapour fluxes measured by the same system. All meteorological, GPP,

and LE data are publicly available via the ICOS data portal. The data follow the standard format of ICOS L2 ecosystem products and are fully compatible with FLUXNET2015. Data processing was performed using the ONEFlux pipeline (https://github.com/icos-etc/ONEFlux). Basic site-level statistics and data coverage are reported in Table 1.

Figure 2: Diurnal (a) and seasonal (b) cycles of ozone concentrations at each site. Shaded areas indicate 95% confidence intervals. Site acronyms are defined in Table 1.

## 2.3 Statistical analysis: partial correlations

To investigate the specific impact of O<sub>3</sub> on GPP, we used a partial correlation analysis, which measures the strength of a relationship between two variables while controlling for the effect of one or more other variables. This analysis isolates the effects of O<sub>3</sub> on GPP, independent of key environmental drivers such as air temperature, short-wave radiation, and vapour pressure deficit (VPD). Despite this control, subsetting the dataset remains valuable for examining the residual impacts of O<sub>3</sub> under specific environmental conditions. These subsets—summer months and midday hours—represent periods of peak biological activity and photochemical reactions, and, therefore, potential O<sub>3</sub> damage. For example, during the summer, ozone concentrations and GPP are generally higher, while during midday, radiation and photosynthesis peak, likely increasing O<sub>3</sub> uptake through stomata. Subsetting, therefore, helps reveal context-specific dynamics and whether the impacts of O<sub>3</sub> are amplified under these conditions. We used the Python package *Pingouin* (Vallat, 2018) to perform the partial correlations and compute the correlation coefficients and their corresponding significance levels (p-values). All partial correlations were computed using only the observed flux and meteorological datasets, independent of the model simulations. To assess the relationship between GPP and O<sub>3</sub>, partial correlations were computed under four configurations for each site:

- 1) Using the entire dataset across all seasons.
- 2) Use summer months only (June, July, and August) when O<sub>3</sub> levels are elevated and foliage is fully developed.

- Restricting the analysis to the period between 12:00 and 16:00, coinciding with peak radiation, photosynthesis, and O<sub>3</sub> levels.
- 4) Combining conditions (2) and (3), focusing on summer midday data.

#### 2.4 JULES land surface model

stomatal conductance (g<sub>p</sub>, m s<sup>-1</sup>) is represented as:

178179

185

This study utilises JULES version 7.4, a community land surface model widely applied as both a standalone model and the land surface component of the Met Office Unified Model (<a href="https://jules.jchmr.org/">https://jules.jchmr.org/</a>, last access: 14 July 2024). We employed the offline version of JULES, prescribing in situ observed meteorological, CO<sub>2</sub>, and O<sub>3</sub> datasets as external forcing inputs. Detailed descriptions of JULES can be found in Best et al. (2011), Clark et al. (2011), and Harper et al. (2016). The Farquhar photosynthesis scheme (Farquhar et al., 1980), as implemented by Oliver et al. (2022), models the leaf-level biochemistry of photosynthesis (A, kg C m<sup>-2</sup> s<sup>-1</sup>) for C<sub>3</sub> vegetation, while the Medlyn scheme (Medlyn et al., 2011) is used to calculate stomatal conductance (g<sub>p</sub>, m s<sup>-1</sup>). The Medlyn approach optimises the stomatal aperture to balance carbon gain with water loss. The

$$g_p = 1.6RT_l \frac{A}{c_a - c_i} \tag{1}$$

where the factor 1.6 accounts for the ratio of diffusivities of  $H_2O$  to  $CO_2$  through stomata, converting stomatal conductance from  $CO_2$  to water vapour units  $(g_p)$ , as required for accurately estimating ozone uptake. R is the universal gas constant  $(J \text{ mol}^{-1} \text{ K}^{-1})$ ,  $T_1$  is the leaf surface temperature (K), and  $c_a$  and  $c_i$  (both Pa) are the leaf surface and internal  $CO_2$  partial pressures, respectively. In this scheme,  $c_i$  is calculated as:

$$c_i = c_a \frac{g_1}{g_1 + \sqrt{d_g}} \tag{2}$$

where  $d_q$  is the specific humidity deficit at the leaf surface (kPa), and  $g_1$  (kPa<sup>0.5</sup>) represents the sensitivity of  $g_p$  to the assimilation rate, which is Plant Functional Type (PFT) dependent. Photosynthesis and stomatal conductance are modelled to respond to changes in environmental drivers (temperature, VPD, incoming radiation, CO<sub>2</sub> concentration and water availability). The impact of soil moisture availability on stomatal conductance is modelled using a dimensionless soil water stress factor ( $\beta$ , unitless) related to the actual soil water content in each layer ( $\theta_k$ , m<sup>3</sup> m<sup>-3</sup>) and the critical water content ( $\theta_{crit}$ , m<sup>3</sup> m<sup>-3</sup>) and water contents at the wilting point ( $\theta_{wilt}$ , m<sup>3</sup> m<sup>-3</sup>) and at which the plant starts to become water stressed ( $\theta_{upp}$ , m<sup>3</sup> m<sup>-3</sup>). The  $\theta_{wilt}$  and  $\theta_{crit}$  are derived from soil matric potentials of -1.5 MPa and -0.033 MPa, respectively (Harper et al., 2021):

$$\beta = \begin{cases} \frac{\theta_k - \theta_{wilt,k}}{\theta_{upp,k} - \theta_{wilt,k}} \; \theta_{wilt,k} \le \theta_k \le \theta_{upp,k} \\ 0 \; \theta_k \le \theta_{wilt,k} \end{cases}$$
(3)

 $\theta_{upp}$  is a function of  $\theta_{crit}$ , and  $p_0$  (unitless), a PFT-dependent parameter, a threshold at which the plant starts to experience water

stress:

$$\theta_{upp} = \theta_{wilt} + (\theta_{crit} - \theta_{wilt})(1 - p_0) \tag{4}$$

198

In this study, the soil drought stress factor β is calculated from the model-simulated soil moisture in JULES. This approach ensures internal consistency with the model's soil properties, hydraulic structure, and root zone distribution. Observed soil

- moisture was not used, even where partially available, due to inconsistent quality, limited depth coverage, and lack of
- harmonised measurements across sites.

## 2.4.1 JULES: Ozone damage scheme

- The ozone damage scheme implemented in JULES follows the approach of Sitch et al. (2007), incorporating a damage factor
- (F) to quantify O<sub>3</sub>-induced reductions in photosynthesis and stomatal conductance. The modified equations for photosynthesis
- ( $A_{net}$ ) and stomatal conductance ( $g_s$ ) under  $O_3$  stress are:

$$202 A_{net} = AF (5)$$

$$g_s = g_p F \tag{6}$$

- where A and g<sub>p</sub> are the photosynthesis and the stomatal conductance without O<sub>3</sub> effects, respectively. In JULES, photosynthesis
- and stomatal conductance are first calculated based on standard environmental inputs (e.g. light, temperature, VPD and CO<sub>2</sub>),
- without considering ozone. Ozone damage is then applied as a separate multiplicative reduction based on the instantaneous
- stomatal ozone flux. The damage factor is given by:

$$F = 1 - a \max[F_{03} - F_{03}r_{it}, 0]$$
(7)

- where  $F_{O3}$  is the  $O_3$  deposition flux through stomata (nmol m<sup>-2</sup> s<sup>-1</sup>),  $F_{O3}$  erit is the threshold for stomatal  $O_3$  uptake (nmol m<sup>-2</sup> s<sup>-1</sup>).
- ¹), and 'a' is the gradient of the O<sub>3</sub> dose-response function (nmol<sup>-1</sup> m<sup>2</sup> s). Both a and FO3<sub>crit</sub> are plant functional type (PFT)
- specific parameters (Table 2). The parameter a determines the slope of the ozone dose-response function and represents how
- sensitive photosynthesis and stomatal conductance are to O<sub>3</sub> uptake. In JULES, a has two default values for each PFT,
- corresponding to "high" and "low" sensitivities to ozone. These two values allow for the exploration of variability in plant
- responses to ozone stress, providing a range of potential outcomes. The flux of  $O_3$  to the stomata  $(F_{O_3})$  is modelled using a flux
- gradient approach:

$$F_{03} = \frac{[o_3]}{r_a + \frac{k_{03}}{g_s}} \tag{8}$$

- where  $[O_3]$  is the molar concentration of  $O_3$  above the canopy (nmol m<sup>-3</sup>),  $r_a$  is the aerodynamic and boundary layer resistance
- (s m<sup>-1</sup>) and  $k_{O3} = 1.67$  (dimensionless) accounts for the relative difference in diffusivities of  $O_3$  and  $H_2O$  through leaf stomata.

#### 2.4.2 Calibration of JULES with and without ozone

In this study, we applied an optimisation approach to calibrate the photosynthesis and stomatal conductance modules in JULES for each site using flux tower datasets. This calibration was performed at a half-hourly resolution, ensuring the optimisation captures short-term variability in GPP responses to environmental drivers. We focused on the summer months (June to August) when O<sub>3</sub> concentrations are typically higher (Table 1, Figure 2), leaves are fully developed, and phenological effects that strongly influence seasonal GPP trends are minimised. At each site, 70% of the available GPP and meteorological data were randomly selected for model calibration, with the remaining 30% reserved for independent validation. This random sampling was applied across the observational period (see Table 1), ensuring both subsets captured a representative range of seasonal and interannual variability.

## **Optimisation** approach

236

249

We employed a two-step calibration approach, conducting separate simulations with and without O<sub>3</sub> effects. We used the Limited-memory Broyden-Fletcher-Goldfarb-Shanno with bound constraints (L-BFGS-B) algorithm (Liu and Nocedal, 1989). This computationally efficient method approximates the Hessian using a subset of past gradients. This makes it particularly suitable for optimising a large number of parameters under bound constraints. The objective function was the Root Mean Square Error (RMSE) between observed and modelled GPP. Optimisation was implemented in Python using the scipy optimize minimize interface and coupled to JULES via scripted automation. Simulations were monitored using cylc scan to ensure successful completion. Convergence was defined as either an RMSE change  $< 1 \times 10^{-10}$  or a maximum of 1000 iterations, Initial values were drawn from JULES defaults (Table 2), and parameter-specific lower and upper bounds were defined based on plausible biophysical ranges (Table S1). The full list of optimised parameters and their boundaries is provided in Table S1. All parameter trajectories, RMSE values, and convergence diagnostics were robustly logged. A safeguard mechanism was included to prevent runaway iteration or crashes due to I/O interruptions.

## Step 1: Optimisation without O<sub>3</sub> effects

- For the simulations without O<sub>3</sub>, we optimised a total of five physiological parameters related to stomatal conductance, 242 photosynthesis, and plant water stress response (Table 2):
  - 1. g<sub>1</sub>: a parameter related to the stomatal conductance model, which determines the sensitivity of stomatal conductance to the assimilation rate.
  - 2. Three photosynthetic parameters:
    - J<sub>max</sub>: V<sub>cmax</sub>: the ratio of the maximum potential electron transport rate at 25°C (J<sub>max</sub>) to Rubisco's maximum rate of carboxylation at 25°C (V<sub>cmax</sub>).
    - i<sub>v</sub> and s<sub>v</sub>: the intercept and slope of the linear relationship between Vc<sub>max</sub> and N<sub>a</sub>, the leaf nitrogen per unit area:

$$V_{cmax} = i_v + s_v N_a \tag{7}$$

Where N<sub>a</sub> is calculated as the product of the Leaf mass per unit area and the top-leaf nitrogen concentration.

3. p<sub>0</sub>: a parameter describing the plant transpiration response to soil moisture, representing the threshold at which the plant begins to experience drought stress.

## Step 2: Optimisation with O<sub>3</sub> effects

- For simulations with O<sub>3</sub>, we extended the optimisation to include two additional ozone-specific parameters:
  - 1. FO3<sub>crit</sub>: the critical flux of O<sub>3</sub> to vegetation, representing the threshold above which O<sub>3</sub> begins to damage photosynthesis and stomatal conductance.
  - 2. a: an empirical PFT-specific O<sub>3</sub> sensitivity parameter that determines the slope of the O<sub>3</sub> dose-response function.
- The optimisation process for simulations with O<sub>3</sub> involved two steps:
  - 1. **Initial optimisation**: The same five physiological parameters as in the no-O<sub>3</sub> simulations were optimised, along with  $FO3_{crit}$  and a.
  - 2. **Local refinement**: To further improve model accuracy under O<sub>3</sub> stress conditions, we performed a local refinement of FO3<sub>crit</sub> and a. Using the optimised parameter set from the initial step, we systematically explored a fine grid of values around the best-performing FO3<sub>crit</sub> and a. Step sizes ranging from 0.005 to 0.025 were used to refine the parameter estimates. Model performance was evaluated for each simulation using RMSE, and the best parameter set was selected based on its agreement with observed half-hourly GPP values.
- In total, we considered four model configurations (Table 3): default and optimised simulations, each with and without ozone. For each site, this setup enabled a direct comparison of model skill under default and optimised parameter sets, as well as the mechanistic contribution of ozone effects.

Table 2: Default parameter values of the JULES for each site.

| Parameter                           | Name         | Unit   | FI-Hyy | FI-Var | BE-Bra | FR-Fon | IT-BFt | IT-Cp2 |
|-------------------------------------|--------------|--------|--------|--------|--------|--------|--------|--------|
| $\mathbf{g_1}$                      | Sensitivity  | kPa0.5 | 2.35   | 2.35   | 2.35   | 4.45   | 4.45   | 3.37   |
|                                     | of the       |        |        |        |        |        |        |        |
|                                     | stomatal     |        |        |        |        |        |        |        |
|                                     | conductance  |        |        |        |        |        |        |        |
|                                     | to the       |        |        |        |        |        |        |        |
|                                     | assimilation |        |        |        |        |        |        |        |
|                                     | rate         |        |        |        |        |        |        |        |
| J <sub>max</sub> :V <sub>cmax</sub> | Ratio of     | -      | 1.48   | 1.48   | 1.48   | 1.78   | 1.78   | 1.63   |
|                                     | Jmax to      |        |        |        |        |        |        |        |
|                                     | Vcmax at 25  |        |        |        |        |        |        |        |
|                                     | deg C        |        |        |        |        |        |        |        |
| i <sub>v</sub>                      | Intercept of | μmol   | 6.32   | 6.32   | 6.32   | 5.73   | 5.73   | 3.90   |
|                                     | the linear   | CO2 m- |        |        |        |        |        |        |
|                                     | relationship | 2 s-1  |        |        |        |        |        |        |
|                                     | between      |        |        |        |        |        |        |        |

|                     | Vcmax and<br>Na                                                      |                             |       |       |       |       |       |       |
|---------------------|----------------------------------------------------------------------|-----------------------------|-------|-------|-------|-------|-------|-------|
| S <sub>v</sub>      | Slope of the<br>linear<br>relationship<br>between<br>Vcmax and<br>Na | μmol<br>CO2<br>gN-1 s-<br>1 | 18.15 | 18.15 | 18.15 | 29.81 | 29.81 | 28.40 |
| p <sub>0</sub>      | threshold at which the plant starts to experience water stress       | -                           | 0     | 0     | 0     | 0     | 0     | 0     |
| FO3 <sub>crit</sub> | Critical flux of 03 to vegetation                                    | nmol<br>m-2 s-1             | 1.6   | 1.6   | 1.6   | 1.6   | 1.6   | 1.6   |
| "High" a            | PFT-specific<br>03<br>sensitivity<br>parameter                       | nmol-1<br>m2 s              | 0.075 | 0.075 | 0.075 | 0.15  | 0.15  | 0.15  |
| "Low" a             | PFT-specific<br>03<br>sensitivity<br>parameter                       | nmol-1<br>m2 s              | 0.02  | 0.02  | 0.02  | 0.04  | 0.04  | 0.04  |

Table 3: Overview of the two types of simulations considered in this study. Default simulations represent site-level runs with model default parameters with or without  $O_3$  effects. Simulations with optimised parameters are also run with and without  $O_3$  effects. In the optimised simulations without ozone, five parameters were calibrated: the sensitivity of stomatal conductance to the assimilation rate (g<sub>1</sub>), the intercept (i<sub>v</sub>) and the slope (s<sub>v</sub>) of the linear relationship between  $V_{cmax}$  and  $N_a$ , the ratio between the carboxylation rate and the rate of electron transport at 25°C ( $J_{max}$ : $V_{cmax}$ ) and the threshold at which the plant starts to experience drought stress (p<sub>0</sub>). For configurations with  $O_3$ , we also add the critical flux of  $O_3$  to vegetation ( $F_{O3crit}$ ) and PFT-specific  $O_3$  sensitivity parameter (a).

| Configuration                    | O <sub>3</sub> effects | Optimised parameters                                                        |
|----------------------------------|------------------------|-----------------------------------------------------------------------------|
| Default (no O <sub>3</sub> )     | No                     | None                                                                        |
| Default (with O <sub>3</sub> )   | Yes                    | None                                                                        |
| Optimised (no O <sub>3</sub> )   | No                     | $g_1$ , $J_{max}$ : $V_{cmax}$ , $i_v$ , $s_v$ , $p_0$                      |
| Optimised (with 0 <sub>3</sub> ) | Yes                    | $g_1$ , $J_{max}$ : $V_{cmax}$ , $i_v$ , $s_v$ , $p_0$ , $FO3_{crit}$ , $a$ |

#### Model evaluation

In order to evaluate the model performance, JULES was forced with the meteorology, CO<sub>2</sub> and O<sub>3</sub> observed at each site and evaluated against flux GPP data. In all simulations, the vegetation cover was prescribed using JULES default PFTs. In each simulation, phenology was simulated prognostically, allowing the model to simulate the dynamic evolution of the maximum

leaf area index (LAI). Prior to running the simulations, the model underwent a 50-year spin-up phase to ensure that the model state variables were representative of steady-state conditions. We used Root Mean Squared Error (RMSE) and the coefficient of determination (r<sup>2</sup>) to quantify the differences between the outputs from the various model simulations and the observations.

## 2.4.3 High-O<sub>3</sub> days analysis

288

294295

To address our third research question—how ozone impacts interact with other environmental factors and how an optimised model can help elucidate these mechanisms—we focused on days when ambient ozone concentrations exceeded 40 ppb at each site. These high-ozone events typically coincide with elevated solar radiation and vapour pressure deficit (VPD), which can enhance stomatal ozone uptake and intensify physiological stress. For each site, we analysed the diurnal cycles of observed GPP, modelled GPP with and without ozone effects, as well as modelled stomatal conductance, latent heat flux (LE), VPD, the soil moisture stress factor (β), and ozone flux to vegetation (FO<sub>3</sub>). The flux FO<sub>3</sub> represents the actual rate of ozone uptake through stomata, computed in JULES from canopy-level ozone concentrations and stomatal plus aerodynamic resistances. The variable B is a dimensionless scaling factor (ranging from 0 to 1) that modulates stomatal conductance in response to soil moisture availability. It reflects the degree of physiological drought stress as perceived by the plant and is derived from the soil water content and site-specific hydraulic thresholds (e.g. wilting point,  $p_0$ ). We used  $\beta$  instead of raw plant-available soil moisture because it is directly integrated into the stomatal conductance formulation in JULES, ensuring model-consistent representation of water stress. Unlike absolute soil moisture, which varies with soil texture and rooting depth, β normalises water limitation in a physiologically meaningful and site-comparable way. This diagnostic framework enabled us to evaluate how well the optimised model captures dynamic interactions between ozone exposure and environmental stressors during high-risk periods. We compared observed and simulated GPP responses across different environmental regimes and examined site-specific optimised parameters, including  $g_1$ ,  $p_0$ , a, and  $FO3_{crit}$ . Our aim was to determine whether GPP reductions are primarily driven by (a) stomatal limitation due to drought and/or high VPD, (b) biochemical ozone damage due to high cumulative ozone uptake, or (c) the simultaneous presence of multiple environmental stressors.

The inclusion of modelled stomatal conductance and FO<sub>3</sub> allows direct tracing of ozone uptake, while the soil moisture stress factor β provides a mechanistic indicator of water limitation. This approach supports a process-level understanding of the mechanisms underlying ozone impacts on carbon uptake during extreme conditions.

#### 2.4.4 GPP reductions due to ozone

To quantify the overall impact of  $O_3$  on GPP, we calculated the relative reduction in GPP for each site using the optimised simulations and the configuration without  $O_3$  impact as the baseline. This calculation was performed each year to account for interannual variability, and the results were averaged to obtain the mean relative reduction over the study period. We define forest sensitivity to  $O_3$  as the percentage reduction in mean annual GPP between the optimised simulations with and without ozone effects. Additionally, we use partial correlation coefficients between observed GPP and ozone concentrations, while

controlling for temperature, radiation, and vapour pressure deficit, as a complementary indicator of site-level sensitivity or resilience. Together, these metrics provide a consistent, quantitative framework for classifying sites as either ozone-sensitive or ozone-resilient and are applied throughout the manuscript in both model evaluation and the interpretation of site-specific responses.

#### 3 Results

## 3.1 Statistical Analysis: Partial correlations

The results of the partial correlation analysis highlight varying degrees of GPP sensitivity to ozone across the investigated sites (Fig. 3). Hyytiälä (FI-Hyy), Värriö (FI-Var), Brasschaat (BE-Bra), Fontainebleau-Barbeau (FR-Fon), and Bosco-Fontana (IT-BFt) exhibited consistently negative correlations between GPP and O<sub>3</sub>, indicating a significant vulnerability to ozone pollution. The negative impact of ozone on GPP is particularly pronounced during specific conditions, such as the summer months (June, July, and August) and midday hours when radiation and temperature are high. While partial correlations control for key environmental variables such as temperature, radiation, and VPD, subsetting the dataset allows for an investigation of the residual impacts of O<sub>3</sub> under specific ecological conditions. These subsets, such as summer months or midday hours, represent periods of peak biological activity and potential O<sub>3</sub> damage, making them ecologically and practically relevant. O<sub>3</sub> concentrations tend to peak during summer due to enhanced photochemical production from increased solar radiation, higher temperatures, and elevated emissions of ozone precursors (NO<sub>x</sub> and VOCs). While plant activity contributes to biogenic VOC emissions, it also increases ozone deposition via stomatal uptake, leading to complex and site-dependent seasonal patterns. Subsetting ensures the analysis captures O<sub>3</sub> impacts under these seasonal conditions. Similarly, during midday hours, when radiation and photosynthesis peak, O<sub>3</sub> uptake through stomata may also reach its highest levels. This approach allows us to determine whether O<sub>3</sub> impacts are consistent across varying contexts or are amplified under specific conditions of heightened environmental and biological activity. Across the sites, FI-Hyy showed weak but significant negative correlations across all subsets, indicating a mild sensitivity to ozone. FI-Var exhibited slightly stronger negative correlations than FI-Hyy, particularly during midday hours in the summer, emphasising the vulnerability of boreal forest ecosystems to ozone stress under specific conditions. BE-Bra and IT-BFt demonstrated the most pronounced negative correlations during the combined summer and midday subsets, suggesting that these conditions heighten the vulnerability of these sites to ozone pollution. Notably, BE-Bra showed the strongest correlation during the summer midday period, underscoring the importance of environmental stressors in exacerbating ozone effects. FR-Fon also displayed significant negative correlations, although the magnitude was generally lower than at BE-Bra and IT-BFt, indicating a moderate sensitivity to ozone.

Conversely, the Castelporziano 2 (IT-Cp2) site showed a negative correlation when using the full dataset; however, correlations for the subset periods became positive and non-significant. This may be due to the limited data availability for IT-Cp2 and

specific site characteristics, such as partial stomatal closure in response to drought and high VPD during warm seasons. These factors may obscure the direct relationship between ozone and GPP at this Mediterranean site.

Overall, the results emphasise the varying impacts of ozone across different environmental contexts and site-specific conditions. Subsetting the data to account for periods of peak biological activity enhances our understanding of the residual effects of O<sub>3</sub> on GPP after controlling for other critical environmental variables. This nuanced approach offers valuable insights into the dynamics of ozone stress across various European forest ecosystems.

Figure 3: Partial correlation coefficients (unitless) between GPP and  $O_3$  – after controlling for air temperature, short-wave radiation and vapour pressure deficit. The calculations were performed for all datasets (salmon bars), including summer only (blue bars, June, July, and August), midday only (green bars, 12-16H), and midday summer only (purple bars, combined). The significance levels: p-value < 0.001 \*\*\*, p-value < 0.01 \*\*, p-value < 0.05 \*, non-significant (ns).

#### 3.2 JULES GPP simulations

The default JULES model configuration (default parameters, Table 4 and Fig. 4) generally exhibits higher variability and larger deviations from observed GPP values across all sites. The optimisation significantly improves model performance by reducing RMSE and increasing r<sup>2</sup> values across most sites (Table 4). However, the incorporation of O<sub>3</sub> effects yields mixed results, with improvements in RMSE at certain sites (e.g., FR-Fon, IT-BFt) but little to no improvement at others, such as FI-Hyy and BE-Bra (Table 4).

- At FI-Hyy, both default and optimised models perform well, with slight improvements in RMSE and r<sup>2</sup> following optimisation.
- The optimised simulation with O<sub>3</sub> achieves the greatest reduction in RMSE (2.11 μmol CO<sub>2</sub> m<sup>-2</sup> s<sup>-1</sup>), a 27% decrease relative
- to the optimised no O<sub>3</sub> case (2.88 µmol CO<sub>2</sub> m<sup>-2</sup> s<sup>-1</sup>), and an increase in r<sup>2</sup> to 0.86 (+3.6%). These improvements reflect the
- model's ability to adjust to local conditions with minimal parameter changes (Fig. 6), particularly in boreal settings. However,
- the inclusion of O<sub>3</sub> does not significantly alter RMSE, suggesting that GPP at this site is not highly sensitive to ozone stress.
- This limited impact is consistent with the relatively low ambient ozone concentrations observed at FI-Hyy, which reduce the
- potential for strong O<sub>3</sub>-induced reductions in GPP.
- At FI-Var, optimisation reduces underestimations during midday peaks and aligns simulated GPP with observations.
- Therefore, the optimised configuration achieves a 1.65 µmol CO<sub>2</sub> m<sup>-2</sup> s<sup>-1</sup> RMSE (-32% relative to 2.41) and a r<sup>2</sup> value of 0.75
- (+2.7%). Key parameter adjustments, such as increases in g<sub>1</sub> and decreases in p<sub>0</sub> (Figs. 5a and 5e), contribute to these
- improvements. Incorporation of O<sub>3</sub> effects only slightly improves RMSE at FI-Var, suggesting moderate sensitivity to ozone
- impacts at this boreal site.
- At BE-Bra, the default configuration performs well, and optimisation further reduces RMSE and improves r<sup>2</sup>. The optimised
- simulation achieves an RMSE of 3.36  $\mu$ mol CO<sub>2</sub> m<sup>-2</sup> s<sup>-1</sup> (-14.3% from 3.92) and an r<sup>2</sup> of 0.81 (+5.2% from 0.77), highlighting
- the importance of fine-tuning parameters such as g1 and sv (Figs. 5a and 5d). However, the inclusion of O<sub>3</sub> has a minimal
- impact on RMSE at this site, suggesting relatively low ozone sensitivity compared to other locations.
- At FR-Fon, default simulations significantly underestimate GPP during peak hours, especially under high ozone stress. The
- optimisation improves model accuracy, showing a reduction in RMSE (5.71 μmol CO<sub>2</sub> m<sup>-2</sup> s<sup>-1</sup>, -35% from 8.72) and an
- increase in r<sup>2</sup> (0.60, +22.4% from 0.49). Despite these improvements, some underestimation remains, indicating that additional
- refinement of O<sub>3</sub> response mechanisms or GPP modelling may be needed at this site.
- At IT-BFt, the default model exhibits large variability in GPP, reflecting the challenges of modeling Mediterranean
- ecosystems. The optimised configuration achieves the greatest improvements, reducing RMSE to 3.78 umol CO<sub>2</sub> m<sup>-2</sup> s<sup>-1</sup>
- (-13.1% from 4.35) and increasing r<sup>2</sup> to 0.82 (+9.3% from 0.75). Adjustments of FO3crit, a, and p0 (Fig. 5f, 5g, and 5e)
- enhance performance by accounting for the combined effects of ozone and water stress, which act as co-limiting factors during
- the summer and jointly contribute to reduced GPP at this site.
- At IT-Cp2, the default model underestimates GPP during midday peaks, particularly under ozone stress. The optimised
- configuration achieves the best results, reducing RMSE to 2.85 umol CO<sub>2</sub> m<sup>-2</sup> s<sup>-1</sup> (-22.8% from 3.69) and increasing r<sup>2</sup> to 0.72
- (+2.9% from 0.70). Adjustments to FO3crit and a play a critical role in capturing ozone impacts at this Mediterranean site,
- demonstrating the necessity of refining these parameters in high-ozone environments.
- Overall, parameter optimisation improves model accuracy and reliability across all sites. However, the inclusion of O<sub>3</sub> effects
- leads to site-specific responses, with improvements in RMSE at some sites (e.g., FR-Fon, IT-BFt) but minimal changes in r<sup>2</sup>
- across most locations. Figure 4 highlights that in some cases, simulations including O<sub>3</sub> effects exhibit increased model biases,
- despite RMSE values suggesting only slight degradation in performance. These findings underscore the need for continued

refinement of ozone response mechanisms to improve model accuracy, particularly in Mediterranean regions where ozone exposure and water stress are strongly coupled.

Table 4: Summary of model evaluation metrics: root mean square error (RMSE,  $\mu$ mol CO<sub>2</sub> m<sup>-2</sup> s<sup>-1</sup>) and coefficient of determination (r<sup>2</sup>) values for each site. The metrics are calculated for default and optimised simulations with and without ozone impacts.

|                        | FI-<br>Hyy | FI-<br>Hyy     | FI-<br>Var | FI-<br>Var     | BE-<br>Bra | BE-<br>Bra     | FR-<br>Fon | FR-<br>Fon     | IT-<br>BFt | IT-<br>BFt     | IT-<br>Cp2 | IT-<br>Cp2     |
|------------------------|------------|----------------|------------|----------------|------------|----------------|------------|----------------|------------|----------------|------------|----------------|
| Default                | 1199       |                | 7 412      | 7 (4)          | Diu        | Diu            | 1011       | 1011           | Dit        | Dit            | GP2        | OP-            |
| Metrics                | RMSE       | r <sup>2</sup> |
| Without O <sub>3</sub> | 2.88       | 0.83           | 3.87       | 0.63           | 4.06       | 0.76           | 9.53       | 0.39           | 6.30       | 0.53           | 3.81       | 0.65           |
| With O <sub>3</sub>    | 2.85       | 0.83           | 3.08       | 0.65           | 3.97       | 0.77           | 8.85       | 0.48           | 5.78       | 0.60           | 3.73       | 0.69           |
| Optimised              |            |                |            |                |            |                |            |                |            |                |            |                |
| Metrics                | RMSE       | r <sup>2</sup> |
| Without O <sub>3</sub> | 2.88       | 0.83           | 2.41       | 0.73           | 3.92       | 0.77           | 8.72       | 0.49           | 4.35       | 0.75           | 3.69       | 0.70           |
| With O <sub>3</sub>    | 2.11       | 0.86           | 1.65       | 0.75           | 3.36       | 0.81           | 5.71       | 0.60           | 3.78       | 0.82           | 2.85       | 0.72           |

Figure 4: Comparison of the observed and simulated GPP diurnal cycles across all sites, averaged over the full year: (a) FI-Hyy, (b) FI-Var, (c) BE-Bra, (d) FR-Fon, (e) IT-BFt and (f) IT-Cp2. Shaded areas encompass plus and minus one standard deviation. The black line represents the observed GPP. The default simulated GPP are the dashed purple line (without O<sub>3</sub>) and dashed green line (with O<sub>3</sub>), and optimised simulated GPP are the purple line (without O<sub>3</sub>) and green line (with O<sub>3</sub>).

- Figure 5: Comparison of default and optimised parameters. The figure presents a comparison between the default (salmon bars)
- and optimised parameter values: without ozone (blue bars) and with ozone (green bars) for the six sites. The parameters include (a)
- stomatal conductance sensitivity to assimilation rate (g<sub>1</sub>), (b) the ratio of maximum potential electron transport rate to maximum
- carboxylation rate (J<sub>max</sub>:V<sub>cmax</sub>), (c) and (d) parameters related to leaf nitrogen (i<sub>v</sub> and s<sub>v</sub>), (e) soil moisture stress threshold (p<sub>0</sub>), (f)
- the critical ozone flux (FO3<sub>crit</sub>), and (g) the sensitivity parameter (a).

424

#### 3.3 Interaction of O<sub>3</sub> with environmental factors on GPP during high ozone days

- For high O<sub>3</sub> days (above 40 ppb), across all sites, the observed GPP shows a characteristic peak around midday, with simulated
- GPP that includes O<sub>3</sub> effects generally aligning more closely with the observed data compared to simulations that exclude O<sub>3</sub>
- effects (Fig. 6). However, the magnitude of this improvement varies by site.
- Ozone concentrations follow a diurnal cycle, peaking in the afternoon (12:00–16:00) across all sites. This peak reflects the
- influence of high solar radiation, temperature, and boundary layer dynamics, including the entrainment of ozone-rich free
- tropospheric air masses that contribute to surface ozone enhancement. The impact of O<sub>3</sub> on GPP is modulated by interactions
- with key environmental factors such as VPD, latent heat flux (LE), and soil moisture stress (β), each influencing stomatal
- conductance (g<sub>s</sub>) and thereby ozone uptake (FO<sub>3</sub>). LE reflects evaporative demand and water availability, while β provides a
- direct measure of soil moisture constraint on stomatal opening. FO<sub>3</sub> represents the actual flux of ozone into the leaves via
  - stomata, and g<sub>s</sub> integrates the stomatal response to multiple environmental drivers, including VPD and soil water availability.
- Around midday, when VPD and LE typically peak, stomatal conductance may decline as a protective response to water loss.
- However, the simultaneous increase in radiation and temperature can elevate ambient O<sub>3</sub> concentrations and photosynthetic
- demand. These competing environmental influences affect O<sub>3</sub> uptake and its impact on photosynthesis, depending on site-
- specific conditions and plant water regulation strategies.
- Figure 6 highlights these dynamics using averaged diurnal cycles of GPP, O<sub>3</sub>, VPD, LE, g<sub>8</sub>, FO<sub>3</sub>, and β during high-O<sub>3</sub> days.
- At the two boreal sites (FI-Hyy and FI-Var), ozone peaks reach moderate levels (~46 and 44 ppb, respectively), but their
- impacts on GPP differ (Table 5). FI-Var shows minimal response to ozone, with an RMSE reduction of just 0.9% (from 3.10
- to 3.07 μmol CO<sub>2</sub> m<sup>-2</sup>s<sup>-1</sup>), suggesting low ecological sensitivity. g<sub>s</sub> and FO<sub>3</sub> values remain relatively low throughout the day,
- and β values are near 1, indicating no significant soil moisture limitations or stomatal downregulation. In contrast, FI-Hyy
- exhibits a large RMSE improvement, from 9.97 to 0.52 μmol CO<sub>2</sub> m<sup>-2</sup> s<sup>-1</sup> (a 94.8% reduction), when ozone effects are included.
- However, this performance gain does not reflect sustained biological sensitivity. Instead, it stems from a systematic
- overestimation of GPP by the ozone-free model during high-O<sub>3</sub> episodes. These episodes are rare (see Table 1), but when they
- do occur, the model without ozone consistently overestimates GPP. The inclusion of ozone damage corrects this bias. The
- partial correlation analysis, combined with the limited ambient ozone exposure outside these rare events, supports this
- interpretation. We therefore distinguish between improved model-data agreement due to structural correction and true
- ecological ozone sensitivity, the latter being more clearly limited at FI-Var.

At BE-Bra, GPP reductions due to ozone are more pronounced, with RMSE dropping from 7.57 to 3.09 μmol CO<sub>2</sub> m<sup>-2</sup> s<sup>-1</sup>, a 59.2% improvement when O<sub>3</sub> effects are considered. This improvement highlights the need to include ozone stress in GPP simulations, particularly in temperate forests where stomatal ozone uptake remains substantial. In Figure 6, g<sub>s</sub> and FO<sub>3</sub> both exhibit midday peaks despite elevated VPD, indicating that stomatal conductance is not fully downregulated under higher evaporative demand, thus allowing more ozone to enter the leaf and cause damage. Interestingly, at FR-Fon, while ozone peaks coincide with midday GPP declines, the difference between with and without O<sub>3</sub> simulations is small. This is confirmed by the minor RMSE reduction from 5.60 to 5.47 umol CO<sub>2</sub> m<sup>-2</sup> s<sup>-1</sup> (2.3%), suggesting that other factors, such as phenology or local climate conditions, play a dominant role in regulating GPP at this site, and that actual ozone uptake is likely limited despite ambient concentrations. Mediterranean sites (IT-BFt and IT-Cp2) experience the highest ozone peaks (>60 ppb). At IT-BFt, the JULES-simulated GPP exhibits a pronounced midday decline, particularly in the optimised configuration with ozone effects, indicating a strong response to midday ozone stress. In these simulations, g<sub>s</sub> shows a clear midday drop, while FO<sub>3</sub> remains high during that period, suggesting that ozone uptake still occurs despite partial stomatal closure. However, the observed GPP shows only a slight morning dip and continues increasing into the afternoon. This divergence points to a potential overestimation of midday stomatal limitation or ozone effects in the model. At IT-Cp2, no distinct midday depression is observed in either the simulated or partitioned GPP. FO<sub>3</sub> is modest, and β remains close to 1 throughout the day, indicating minimal water stress and limited ozone uptake. While these sites do show noticeable reductions in RMSE after including ozone effects, 0.8% at IT-BFt (from 5.88 to 5.83 umol  $CO_2$  m<sup>-2</sup> s<sup>-1</sup>) and 64.6% at IT-Cp2 (from 5.45 to 1.93 umol  $CO_2$  m<sup>-2</sup> s<sup>-1</sup>), these improvements are not the largest among all sites. Indeed, FI-Hyy and BE-Bra show greater RMSE reductions during high ozone days. This suggests that while Mediterranean sites face high ozone concentrations, the degree of ozone-induced GPP reduction may vary depending on the interplay of environmental stressors and model representation. The results highlight the importance of site-specific calibration and caution against generalising Mediterranean sites as the most ozone-sensitive solely based on ozone concentration levels. Interestingly, although JULES simulates strong midday GPP declines at Mediterranean sites, Figure 5 shows that the ozone sensitivity parameters are generally lower for Mediterranean forests. This pattern may reflect the fact that high VPD and limited soil moisture in these regions reduce stomatal conductance during midday, thereby lowering actual ozone uptake and mitigating its physiological effects, despite high ambient O<sub>3</sub> concentrations. This dynamic, documented in several previous studies (Lee et al., 2013), suggests that the observed midday GPP reduction may be driven more by water stress than by direct ozone damage. At IT-BFt, the JULES-simulated GPP exhibits a sharp midday reduction. especially when ozone effects are included, suggesting a modelled compound stress due to high VPD and ozone uptake. However, the partitioned GPP at this site increases during the same period (after 10:00), indicating that stomatal closure due to VPD is not occurring to the extent the model assumes. This divergence suggests a potential overestimation of midday water limitation in the model configuration. At IT-Cp2, neither the modelled nor observed GPP shows a distinct midday dip, indicating that ozone and VPD effects are less pronounced or not synchronised enough to produce a compound stress response. These site-specific dynamics reinforce the need for a more accurate representation of stomatal regulation under co-occurring stresses in Mediterranean systems.

In addition to evaluating RMSE and r², we examined residuals between observed and simulated GPP to identify systematic biases. At several sites, such as IT-BFt, residuals indicated that modelled GPP tended to underestimate peak values during high O₃ periods, particularly around midday. This aligns with the observed mismatch in diurnal dynamics (Fig. 6), suggesting that while optimisation improves overall fit, specific stress responses (e.g. compound O₃ and VPD effects) may still be underestimated or mistimed. These residual diagnostics support the need for further refinement in the representation of ozone damage under variable environmental conditions.

Table 5: Performance of optimised JULES without O<sub>3</sub> and with O<sub>3</sub> for O<sub>3</sub> levels above 40 ppb for each site.

|                     | FI-  | FI-            | FI-  | FI-            | BE-  | BE-            | FR-  | FR-            | IT-  | IT-            | IT-  | IT-            |
|---------------------|------|----------------|------|----------------|------|----------------|------|----------------|------|----------------|------|----------------|
|                     | Hyy  | Hyy            | Var  | Var            | Bra  | Bra            | Fon  | Fon            | BFt  | BFt            | Cp2  | Cp2            |
| Metrics             | RMSE | r <sup>2</sup> |
| Without             | 9.97 | 0.46           | 3.10 | 0.65           | 7.57 | 0.60           | 5.60 | 0.55           | 5.88 | 0.42           | 5.45 | 0.70           |
| $O_3$               |      |                |      |                |      |                |      |                |      |                |      |                |
| With O <sub>3</sub> | 0.52 | 0.85           | 1.18 | 0.70           | 3.09 | 0.73           | 5.47 | 0.59           | 2.31 | 0.65           | 1.93 | 0.77           |

Figure 6: Averaged diurnal cycles of gross primary production (GPP), ozone (O<sub>3</sub>), vapour pressure deficit (VPD), latent heat flux (LE), stomatal conductance (g<sub>s</sub>), ozone flux into leaves (FO<sub>3</sub>), and the soil moisture limitation factor (β) across high ozone days (O<sub>3</sub> above 40 ppb) at six forest sites: (a-c) FI-Hyy, (d-f) FI-Var, (g-i) BE-Bra, (j-l) FR-Fon, (m-o) IT-BFt, and (p-r) IT-Cp2. The left panels show observed GPP (black) and simulated GPP from the optimised model without O<sub>3</sub> (purple) and with O<sub>3</sub> (green), along with ozone concentrations (blue dashed line). The middle panels show VPD (olive) and LE (magenta). The right panels show g<sub>s</sub> (orange), FO<sub>3</sub> (brown), and β (dark slate grey).

#### 3.4 GPP reductions due to O<sub>3</sub> effects

The mean annual GPP reduction varies significantly across the sites, suggesting a site-specific exposure and response to ozone stress (Fig. 7). The negative values indicate a decrease in GPP, highlighting the impact of ozone as a stressor on plant productivity.

FI-Hyy and FI-Var show relatively small reductions in GPP, with annual mean decreases of -1.36 % and -1.04 %, respectively. This suggests that these northern sites are less sensitive to ozone stress, possibly due to lower background O<sub>3</sub> concentrations (Fig. 2, Table 1) or lower stomatal ozone uptake, which limits the damaging effects on GPP. In contrast, IT-BFt and IT-Cp2 exhibit the highest reductions (-6.2% and -5.4%, respectively), which can be attributed to higher ozone exposure (Fig. 2) and greater ozone uptake, exacerbating stress on photosynthesis and stomatal function. Similarly, temperate forests (BE-Bra and FR-Fon) exhibit moderate reductions in GPP, with declines of -5.22% and -2.62%, respectively. While ozone effects at FR-Fon are lower than those at BE-Bra, they are still significant, underscoring that broadleaf deciduous forests also experience ozone-induced productivity losses. The stronger impact at BE-Bra may be linked to higher stomatal ozone uptake, as suggested by the site's parameter sensitivity (Fig. 6).

These findings highlight the need for region-specific ozone mitigation strategies, particularly in Mediterranean ecosystems where ozone-induced reductions in GPP exceed -5 % annually. The combination of high ozone, VPD, and water stress in these regions may further amplify productivity losses, making them particularly vulnerable to future climate and air quality changes.

Figure 7: Annual mean GPP reduction due to ozone exposure (%). The bar plot represents the annual mean reduction in Gross Primary Productivity (GPP) as a percentage for each site: FI-Hyy, FI-Var, BE-Bra, FR-Fon, IT-BFt, and IT-Cp2.

#### 4 Discussion and Conclusions

This study underscores the importance of incorporating ozone effects into the JULES model to enhance its accuracy in simulating Gross Primary Productivity (GPP) across diverse European forest ecosystems. By including ozone effects, the model demonstrated improved performance, particularly during high O<sub>3</sub> events and in central and southern European sites where ozone stress is most pronounced. For example, reductions in RMSE at FR-Fon (from 9.53 to 5.71), IT-BFt (from 6.30 to 3.78), and IT-Cp2 (from 3.81 to 2.85) highlight the significant role of ozone in modulating plant productivity. These findings confirm previous observations that ozone exposure strongly influences plant photosynthesis and carbon sequestration, particularly in Mediterranean climates (Sitch et al., 2007). However, the minimal differences in northern European sites (FI-Hvv and FI-Var) suggest boreal forests' lower sensitivity to ozone, aligning with prior research showing lower ozone uptake in cooler, high-latitude environments (Wittig et al., 2009). The annual mean GPP reductions due to ozone exposure reveal a clear spatial gradient, with northern sites showing minimal reductions (-1.04% to -1.36%) and southern sites experiencing more pronounced decreases (-5.4% to -6.2%). This gradient reflects the interplay of higher ambient ozone concentrations, greater stomatal conductance, and compounding environmental stressors such as high temperatures and vapor pressure deficit in Mediterranean climates (Proietti et al., 2016). Central European sites (e.g., BE-Bra and FR-Fon) exhibited intermediate reductions, consistent with transitional climatic conditions that modulate ozone impacts. These patterns emphasise the importance of considering regional climatic variables in modeling ozone effects on GPP. Although Mediterranean species may possess physiological adaptations to mitigate ozone stress, such as conservative stomatal behaviour, these mechanisms may be insufficient under conditions of sustained high ozone and environmental stress. A key insight from our study is the potential overestimation of ozone impacts in prior modeling efforts. For example, Anav et al. (2011) estimated a 22% reduction in annual GPP across Europe using the ORCHIDEE model, while Oliver et al. (2018) simulated that GPP was reduced by 10 to 20% in temperate regions and by 2 to 8% in boreal regions using JULES. These discrepancies likely stem from differences in the resolution and accuracy of ozone and GPP datasets. By integrating highresolution in situ ozone, meteorology, and GPP measurements, our study provides more precise estimates, reducing the biases

inherent in purely simulation-based approaches. For instance, Gerosa et al. (2022) reported GPP reductions of 2.93% to 6.98% at IT-BFt using statistical models based on in situ data, aligning closely with our findings of a -6.2% GPP reduction. Similar conclusions were found by Conte et al. (2021), who adopted statistical models based on dynamic seasonal thresholds of ozone doses to reduce the bias between observed and modelled GPP. These results highlight the critical role of empirical data in refining model predictions.

This study's diurnal GPP, ozone, VPD, and LE patterns provide additional insights into the interaction between ozone and environmental stressors. Across all sites, ozone concentrations peaked in the late afternoon, coinciding with periods of high VPD and LE. This temporal alignment highlights the role of atmospheric conditions, including high solar radiation and temperatures, in driving ozone formation and stomatal ozone uptake. Southern sites, such as IT-BFt, exhibited a pronounced midday decline in simulated GPP, reflecting modelled ozone sensitivity and the interacting influence of high ozone concentrations and elevated VPD. However, the partitioned GPP at this site does not exhibit the same midday depression; instead, it increases gradually into the afternoon, At IT-Cp2, no midday dip is observed in either the simulated or observed GPP. These findings align with the work of Ainsworth et al. (2012), who demonstrated that multiple stressors can exacerbate the physiological impacts of ozone on plants. At BE-Bra, however, we observed a negative partial correlation between GPP and ozone (Section 3.1). Yet, the inclusion of ozone effects in the model resulted in only modest performance improvements. This contrast may arise from differences in timescale and model sensitivity. Verryckt et al. (2017), who conducted a detailed study at the same Scots pine stand, found no significant long-term GPP reduction attributable to ozone, despite frequent exceedance of critical exposure thresholds such as AOT40 and POD1. However, their residual analysis suggested short-term GPP reductions of up to 16% following days with high stomatal O<sub>3</sub> uptake, particularly in late spring and early summer. These results support the idea that ozone effects at BE-Bra may be episodic and confounded by co-occurring environmental stressors. such as light and temperature. The modest RMSE reduction in our simulations may thus reflect a structural limitation of the JULES damage formulation in capturing such short-lived physiological responses under temperate, humid conditions. In contrast, boreal sites such as FI-Hyy exhibited minimal midday GPP reductions, consistent with their relative resilience to ozone stress under cooler atmospheric conditions. This supports prior research suggesting that boreal species often operate under a narrower range of stomatal conductance, limiting ozone uptake even during peak stress periods (Hoshika et al., 2013, Rannik et al., 2012). The variability in ozone impacts across sites emphasises the need for regional calibration of land surface models like JULES. This study optimised key parameters, including the critical ozone flux, stomatal conductance sensitivity. and ozone sensitivity coefficient, to improve model performance. While the results of this study are specific to the JULES model framework and the six European forest sites, some spatial trends, such as increasing ozone sensitivity (a) and decreasing critical ozone flux thresholds (FO3crit) toward southern latitudes, may reflect broader physiological adaptations to environmental stress gradients. These patterns could inform the understanding of ozone responses in other forest ecosystems with comparable climatic and ecological conditions. However, we explicitly caution against the direct application of these sitecalibrated parameter values to other regions without local validation, as species traits, soil properties, and climatic variability shape ozone responses. Notably, several of the physiological parameters optimised in this study, such as stomatal sensitivity  $(g_1)$ , the photosynthetic capacity ratio  $(J_{max}: Vc_{max})$ , and the soil moisture stress threshold  $(p_0)$ , are shared across multiple land surface and ecosystem models. This overlap suggests broader relevance, but these parameters must still be used with caution, as their values and effects can vary depending on the model structure. Although the quantitative results are JULES-specific. the methodological approach, site-level optimisation using in situ ozone and GPP data with a stomatal flux-based damage formulation, is transferable and could improve ozone-vegetation representation in other modelling frameworks. The JULES ozone damage scheme, as applied in this study, uses the instantaneous stomatal flux of ozone to compute a damage factor (F) that is applied equally to net photosynthesis (A) and stomatal conductance (gp). This approach enables a simple and efficient integration into the model but may not fully capture the temporal dynamics of ozone-induced damage. Many other

546

modeling frameworks use cumulative ozone uptake metrics—such as the phytotoxic ozone dose above a threshold (POD6) to represent damage accumulation over time (Wittig et al., 2007; Lombardozzi et al., 2013). Moreover, empirical evidence indicates that A and g<sub>p</sub> may respond differently to ozone, with distinct sensitivities and temporal responses (Lombardozzi et al., 2012a,b). Future versions of JULES could benefit from decoupling these effects by estimating separate sensitivity parameters (a) and critical thresholds (FO3<sub>crit</sub>) for A and g<sub>p</sub>, and by transitioning toward cumulative flux-based ozone stress formulations.

Our study highlights the importance of integrating long-term in situ measurements into land surface models to improve their accuracy and reliability. Expanding such measurements' spatial and temporal coverage is essential for capturing the full variability of ozone impacts across biomes and climatic conditions. Future research should also prioritise refining ozone response mechanisms in land surface models, particularly in regions where multiple stressors interact to influence plant productivity. For example, incorporating dynamic responses to heat waves, droughts, and other extreme events could provide a more comprehensive understanding of how ozone stress interacts with climate change.

#### **Code Availability**

JULES-vn7.4 was used for all simulations. The JULES model code and suite used to run the model are available from the Met Office Science Repository Service (MOSRS). Registration is required, and the code is available to anyone for non-commercial use (for details of licensing, see https://jules.jchmr.org/code, last access: 29 June 2024). Visit the JULES website (https://jules.jchmr.org/getting-started, last access: 29 June 2024) to register for a MOSRS account. Documentation for the JULES model is located at https://jules-lsm.github.io/vn7.4/ (last access: 29 June 2024). Site-level simulations used the rose suite u-dg903 (https://code.metoffice.gov.uk/trac/roses-u/browser/d/g/9/0/3, at revision 289677), which is a copy of the ual752 **JULES** suite for **FLUXNET** 2015 and LBA sites described https://code.metoffice.gov.uk/trac/jules/wiki/FluxnetandLbaSites (last access: 29 June 2024) and downloaded from https://code.metoffice.gov.uk/trac/roses-u/browser/a/1/7/5/2/ (Harper et al., 2021) at revision 286601. Suites can be downloaded from MOSRS once the user has registered for an account.

#### Data Availability

The ICOS data (meteorological variables, fluxes and carbon dioxide concentration) used to run JULES are available for download from <a href="https://www.icos-cp.eu/observations">https://www.icos-cp.eu/observations</a> (last access: 29 June 2024). The ozone data was obtained by requesting the PIs of each site, except Värriö, obtained through the SMEAR I research station (Kolari et al., 2024) and Hyytiälä, available on SMEAR II Hyytiälä forest meteorology, greenhouse gases, air quality and soil dataset (Aalto et al., 2023).

#### **Author contribution**

- IV wrote the paper and led the data analysis, with contributions from all authors. FM, SS, FB, PB, MB and HV contributed to
- the interpretation of the data. GG and SF contributed to in situ data for model evaluation. MCDR, SS and FB contributed to
- JULES simulations.

603

#### Competing interests

The authors declare that they have no conflict of interest.

## Acknowledgements

- IV was funded by Fonds Wetenschappelijk Onderzoek Flanders (FWO grant no. G018319N). FM was funded by the FWO as
- a senior postdoc and is thankful to this organisation for its financial support (FWO grant no. 1214723N). SS was funded
- through UKRI NERC funding (NE/R001812/1). MCDR has been supported by the H2020 4C project no. 821003, by the
- NetZeroPlus (NZ+) grant funded by UKRI-BBSRC award BB/V011588/1 and also by the UKRI-'AI for Net Zero' Programme
- project: ADD-TREES, Grant number EP/Y005597/1. The authors acknowledge Ivan Jansens from the University of Antwerp,
- Johan Neirynck from the Instituut Natuur- en Bosonderzoek, and Daniel Berveiller from the University of Paris-Saclay for the
- ozone datasets provided for the Brasschaat and Fontainebleau-Barbeau sites, respectively.

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

- 850
- 851