# Peer review of "Modeling impacts of ozone on gross primary production across"

_EGUsphere, 2025_

## Referee Comment (RC3)

**Review of bg-2025-1375 "", by Vieira et al. on "Modeling impacts of ozone on gross primary production across European forest ecosystems using JULES"**

The paper presents an analysis of the impact of ozone versus mostly meteorological drivers of the GPP of European forests for sites with contrasting conditions regarding pollution levels and physical drivers of plant productivity. It relies both on an statistical analysis of some long-term (> decades) data on GPP and other meteorological variables as well as application of a state-of-the art DGVM (JULES) set-up in an offline mode and driven by the observations. Both the statistical analysis as well as model experiments are applied aiming to identify/quantify the role of O3 uptake as a stressor besides other stresses imposed on vegetation functioning. Overall, I appreciate the followed approach but have some major issues with some specific features of the paper. I agree with the other referees that the last main research question is not really addressed. Disentangling what at the end explains the different responses for the different sites, does not come out well out of this study. I also have some major issues with the descriptions of the role of water stress in the overall response of the vegetation to O3 and other stress terms. There is the reference to the role of the VPD effect on stomatal closure and, consequently, on the O3 effect, but then there is also quite some references that LE also plays a role here. See my specific comments below for further details about this. But what I am missing here is the role of soil water limitation. It is excluded from the data-analysis but also referenced in some inconsistent manner (water stress..) whereas this stress term might be especially relevant for modulating stomatal opening (and photosynthesis?) on longer (weekly/seasonal) timescales and where the VPD is mainly impacting the diurnal cycle. Referring to these different timescales of water stress that might exacerbate the impact of O3 exposure, I also miss completely a discussion on how this study informs about the timescale of the effect and impact by O3 on GPP. Finally, the presentation of the tables, figures and equations should be substantially improved. Overall, based on these observations and considerations, I recommend a major revision of this paper but would be keen then to review a revised version of the ms in due time.

**Specific comments:**

Line 47: the statement on the impact on photosysnthesis/GPP and the following statement in line 49 (Therefore…) misses mentioning the main consequences of the reduced GPP/conductance for climate (and thus the main motivation why to consider the O3 impact on ESMs; the impact on atmospheric CO2, water vapor (reduced LE) but also further increasing O3 itself by reduced O3 deposition.

Line 74: referring to studies that aimed to assess the O3 deposition impact on European forests, it would be very much appreciated to have here the reference explicitly listed.

Table 1 comes out quite poorly; am aware it is most about the information shared in that table but this this table should be presented in a more optimal manner.

Line 241 -- Going through the list of meteorological variables in section 2.2 I am missing here soil moisture. Knowing about its important role in inducing water stress on stomatal opening, this is a parameter that should quite obviously be included here.

Interpreting Figure 2a and b on temporal variability in O3, including the 95% confidence interval, but then also seeing the reported maximum O3 values in Table 1, I wonder what values have been used to determine these long-term mean diurnal and seasonal cycles in O3.

Equations 1 & 2: sloppy to present equations like this in a submitted paper for reviewing

Lines 177/178; here the feature of water availability/soil water limitation is introduced and which raises the question how this will be considered; simply using the model simulated soil moisture balance or using the observed soil moisture.

In section 2.4.2 on calibration of JULES it might be relevant to mention the timeframe of the available dataset that has been used for this step of the approach.

Line 227: for the optimization of the stomatal conductance/photosynthesis representation in JULES experiments without the O3 impact, did you then also use data where O3 was indeed so low that you would not expect any significant impact?

Line 275; upon checking the optimization based on minimizing the RMSE and also checking the impact on r-squared did you also conduct a key check of this optimization approach; checking the residuals? I am curious to see how this comes back in reading further through the results/discussions.

Line 296: In explaining the feature of subsetting it is interesting to read that you state that O3 is higher in summer because of increased plant activity. I don't agree with this statement; there is then also more deposition and which would lower O3 levels. You could be hinting at the role of biogenic VOC and NO emissions being higher but the impact of the VOCs also depends on the mixture of VOCs being emitted.

Line 330: I have been going a couple of times through the following statement: "*The optimised simulation with O3 achieves the greatest reduction in RMSE (2.11 µmol CO$_2$ m$^{-2}$ s$^{-1}$) and an increase in r² (0.86). These improvements reflect the model's ability to adjust to local conditions with minimal parameter changes (Fig. 6), particularly in boreal settings. However, the inclusion of O$_3$ does not significantly alter RMSE, suggesting that GPP at this site is not highly sensitive to ozone stress*". You seem to contradict yourself. I thought you wanted to express that the initial step of optimization of the model, on the settings of calculation of assimilation and conductance, results in a major decrease in RMSE but that then adding the O3 impact does not substantially further decrease the RMSE. But then checking Figure 5 for Hyytiala, the default model without the O3 impact seems to perform quite well and including the O3 impact makes it perform worse. I am getting confused here. Rephrase to make this more clear.

Again, the overall presentation of the tables and figures, like Figure 5, is quite poor. I would suggest to, for example, present the observed GPP line as the reference line, much thicker.

Line 340: in your discussion on the results for the Braschaat site, the model application at the end indicates a low sensitivity to O3, which seems to contradict the initial analysis presented in Section 3.1 for this site suggesting a large impact of O3. This might come back in the discussions (also given the results by the Verryckt 2017 study) but might be good to already shortly reflect on this here.

Line 344: "achieved a 1.65 µmol $CO_2$ $m^{-2}$ $s^{-1}$ RMSE and 0.75 $r^2$", bad english according to me, what is a 0.75 r2?? an r2 value of 0.75.....

Section 3.3; line 378, you discuss on the role of processes explaining the peak in O3 in the afternoon and here mentioning atmospheric dynamics as one of those processes; you could be here more specific referring to the role of atmospheric boundary layer dynamics with the role of entrainment of FT air masses that generally explain to a large extent these peak afternoon values with this entrainment partly compensating for the efficient removal of O3 by surface deposition.

Then in the following line I miss completely the mentioning of the role of soil moisture. You refer here to LE as a parameter influencing stomatal conductance; This is according to me a complete misperception; The LE actually depends on stomatal opening and the available water expressed by the water potential height and which depends strongly on soil water availability.

Line 385: I am getting lost again wrt the results for the Hyytiala site: "*At FI-Hyy, however, simulations without $O_3$ significantly underestimate GPP, leading to a high RMSE (9.97 µmol $CO_2$ $m^{-2}$ $s^{-1}$), which improves dramatically when $O_3$ effects are included (RMSE = 0.52 µmol $CO_2$ $m^{-2}$ $s^{-1}$). This suggests that while FI-Hyy is less sensitive to $O_3$ overall, proper parameterisation of $O_3$ effects improves model performance*". Going back to Figure 5, I see some different behaviour or am I missing here something. And how to reconcile the finding that inclusion of the O3 effect in the model results in such large decrease in RMSE with the notation that at Hyytiala the overall sensitivity of the forest to O3 should be small. Is the optimized model including the O3 impact getting the right results for the wrong reasons?

Line 398: on the findings for the Mediterranean sites there is another interesting statement; "high VPD and stomatal conductance increase O3 uptake"; according to me the high VPD actually results in a strong decrease in stomatal conductance and which decreases the O3 uptake (and impact).

Line 400: "*Interestingly, despite the strong midday declines in GPP at Mediterranean sites, Figure 6 suggests that the ozone sensitivity parameters are generally lower in Mediterranean forests*". This statement suggests a major misperception according to me: the strong midday declines in GPP for those sites, due to the VPD effect (and

potentially further exacerbated by the role of limited soil moisture), might make the vegetation less sensitive to the O3 impact; when the O3 fluxes would be highest due to maximum O3 levels and maximum stomatal opening, the moisture limitation impact actually strongly reduces the impact of O3. This has already been presented in quite many previous studies.

Line 429: here the term water stress comes up again as a main term impacting GPP but so far in the presented analysis, there has not been any further support from the data and model analysis that indicates how important this feature is for the various sites.

Line 446; here it is suggested that higher stomatal uptake (conductances and O3) might explain a larger impact at the more southern sites but have also not seen here any supporting information.

Line 449: "*For instance, Mediterranean species often exhibit adaptations such as enhanced antioxidant production to mitigate ozone damage, though these defenses can be overwhelmed under extreme environmental stress*". This is quite interesting but also strong statement that needs further clarification and, potentially support by references. Are you referring here to specific VOC emissions with the emitted species being very reactive with O3 and which, consequently, reduces the stomatal uptake by the enhanced non-stomatal removal, or are you referring here to other (inside leaf/needle tissues) chemical interactions??

Line 463: Here the following line makes some things clear that actually triggered some of my previous comments: "*Across all sites, ozone concentrations peaked in the late afternoon, coinciding with periods of high VPD and LE*". It makes clear that you used the observations of high LE to infer that also then the stomatal conductance must have been high, despite the high VPD effect. Making this clear at an earlier stage would avoid some of the criticism that I have shared so far.

But then in line 466 I am getting confused again: "*reflecting their heightened sensitivity to ozone and the compounding effects of high VPD and LE*"; First of all, I have honestly not seen strong evidence that the afternoon decrease in GPP for the EU southern sites is really due to the O3 effect. Can it not be mostly the impact of the VPD? And what is the effect of a high LE? A high LE indicates still quite high stomatal conductance despite the high VPD effect. I don't follow this reasoning.

Finally, in your discussion/conclusion section I was awaiting a discussion on the conflicting results on the Braschaat site. The study by Verryckte (2017) indicated that there was no O3 effect to be detected in a long-term data set analysis. Your study gets different results but dependent on if you indeed do the data-analysis (3.1) or the model-based evaluation of the impact. This definitely deserves some more discussion on how to reconcile these contrasting findings.

---

## Author Comment (AC2)

**Author Comments – Response to Referee #2 (RC2)**

We thank Referee #2 for their detailed, thoughtful, and constructive feedback. We are pleased that the reviewer found our manuscript well-written and a significant contribution. Below, we address each of the general, specific, and technical comments.

**General Comments**

**RC2: The reviewer notes that while Questions 1 and 2 are well addressed, the answer to Question 3 lacks clarity due to the absence of stomatal conductance and $O_3$ uptake flux ($FO_3$).**

**AC:** In response, we have revised the manuscript to more explicitly and quantitatively address research question 3: how ozone impacts interact with other environmental factors, and how an optimised model can help us understand these mechanisms, particularly on high-ozone days. To improve clarity, we have added modelled stomatal conductance, ozone flux to vegetation ($FO_3$), and soil moisture as key prognostic variables in our high-ozone day analysis (Section 2.4.3). These outputs allow us to distinguish between: 1) stomatal limitation, where high VPD and/or low soil moisture reduces conductance and $FO_3$, thus limiting $O_3$ damage, and 2) direct ozone stress, where elevated $FO_3$ and maintained stomatal conductance lead to reductions in GPP through biochemical effects. In Section 3.3, we now interpret observed and simulated GPP patterns on high-ozone days using these additional variables to identify the dominant mechanisms at each site. This mechanistic analysis is supported by optimised parameter values (e.g. $g_1$, $p_0$, $a$, and $FO_3crit$), and improves the attribution of GPP reductions to specific environmental and physiological drivers. We believe these additions now provide a clear and complete response to Research Question 3, and we thank the reviewer for prompting this improvement.

**Specific Comments**

**RC2: Since the partitioned GPP is central to the inference made in this manuscript, it would help if the authors offered a description of how GPP was partitioned from observed net carbon flux. This could be as simple as a brief description with a reference to a citation that details the methods. Lines 130 – 132 claim that GPP and LE were estimated from net-carbon flux. Net carbon flux is used to estimate net ecosystem exchange and GPP. LE is not estimated from net carbon flux. It is typically estimated from H2O flux. The authors should consider correcting or clarifying if they have developed a technique or used an existing technique to estimate LE from net carbon flux.**

**AC:** We thank the reviewer for pointing this out. We have revised Section 2.2 to clarify that GPP was derived from net ecosystem exchange (NEE) using standard partitioning approaches implemented in the ICOS ONEFlux pipeline. We also corrected the erroneous statement about LE and now clarify that LE is derived from water vapour ($H_2O$) flux measurements, not carbon flux:

"The half-hourly Gross Primary Production (GPP, $\mu$mol m$^{-2}$ s$^{-1}$) and Latent Heat flux (LE, W m$^{-2}$) were derived from eddy covariance measurements at each site. GPP was estimated from net ecosystem exchange (NEE) using standard partitioning techniques implemented in the ICOS ONEFlux processing pipeline (Warm Winter 2020 Team, ICOS Ecosystem Thematic Centre, 2022). LE was derived from water vapour fluxes measured by the same system. All meteorological, GPP, and LE data are publicly available via the ICOS data portal. The data follow the standard format of ICOS L2 ecosystem products and are fully compatible with FLUXNET2015. Data processing was performed using the ONEFlux pipeline (https://github.com/icos-etc/ONEFlux). Basic site-level statistics and data coverage are reported in Table 1."

**RC2: The JULES damage scheme calculated the O3 damage factor, F, as a function of the stomatal flux of O3 (equation 7). It appears that this is the instantaneous stomatal flux of O3. However, the cumulative flux of O3 through stomata is typically used as the damaging quantity (Lombardozzi et al., 2013, Wittig et al., 2007). Many threshold-based O3 damage indicators are based on cumulative exposure or cumulative stomatal dose (i.e.: AOT40 and POD6). The authors could consider elaborating on this in the discussion section of this manuscript by discussing if it would be worthwhile to use cumulative O3 stomatal flux in future optimization studies. The JULES O3 damage factor, F, as it is formulated in the current study appears to be the same damage factor that is applied to both stomatal conductance (gp) and net photosynthesis (A). However, previous research suggests that net photosynthesis and stomatal conductance are differentially impacted by O3 (Lombardozzi et al., 2012a,b). Both quantities might not exhibit the same sensitivity to O3 or might not change at the same rate as a function of O3 uptake (Lombardozzi et al., 2012b). This suggests the use of separate damage factors, sensitivities, and critical O3 levels for stomatal conductance and net photosynthesis. Are a and FO3crit separately estimated for A and gp? These distinctions are important because they might have implications for modeling transpiration in a land surface model if stomatal conductance is involved. The results report only one value for FO3crit and a which implies that the same damage factor is applied to both A and gp. In the discussion portion of the paper, it would be worth discussing the reasoning behind the JULES modeling choices for the specific formulation of O3 stress on gp and A compared to other methods of incorporating damage factors in land surface models (see Lombardozzi et al. 2012a and b who tried various configurations of an O3 damage factor in the community land model).**

**AC:** We thank the reviewer for this thoughtful suggestion. In the revised Discussion section, we clarify that the current JULES implementation uses instantaneous stomatal O$_3$ flux to compute the damage factor F, applied equally to both photosynthesis (A) and stomatal conductance (gp). We acknowledge that alternative approaches—such as those used in the Community Land Model (CLM)—use cumulative O$_3$ uptake (e.g., POD6) as a more biologically realistic indicator of damage (Lombardozzi et al., 2013; Wittig et al., 2007). We now note that incorporating cumulative dose metrics and distinguishing between photosynthetic and stomatal sensitivities (FO$_3$crit, a) could improve the representation of O$_3$ effects in future JULES developments.

"The JULES ozone damage scheme, as applied in this study, uses an instantaneous stomatal flux of ozone to compute a damage factor (F) that is applied equally to net photosynthesis (A) and stomatal conductance (g$_p$). This approach enables a simple and efficient integration into the model but may not fully capture the temporal dynamics of ozone-induced damage. Many other modeling frameworks use cumulative ozone uptake metrics—such as the phytotoxic ozone dose above a threshold (POD6)—to represent damage accumulation over time (Wittig et al., 2007; Lombardozzi et al., 2013). Moreover,

empirical evidence indicates that A and $g_p$ may respond differently to ozone, with distinct sensitivities and temporal responses (Lombardozzi et al., 2012a,b). Future versions of JULES could benefit from decoupling these effects by estimating separate sensitivity parameters ($a$) and critical thresholds ($FO3_{crit}$) for A and $g_p$, and by transitioning toward cumulative flux-based ozone stress formulations."

**RC2: The diurnal cycles of partitioned and JULES simulated GPP are shown in Figure 5. Can the authors clarify whether these diurnal cycles were estimated using data and simulations from all seasons or just the summer?**

**AC:** The diurnal cycles shown in Figure 4 are based on data and simulations from the full year, not limited to the summer season. We will clarify this in the caption of Figure 4:

"Figure 4: Comparison of the observed and simulated GPP diurnal cycles across all sites, averaged over the full year: (a) FI-Hyy, (b) FI-Var, (c) BE-Bra, (d) FR-Fon, (e) IT-BFt and (f) IT-Cp2. Shaded areas encompass plus and minus one standard deviation. The black line represents the observed GPP. The default simulated GPP are the dashed purple line (without $O_3$) and dashed green line (with $O_3$), and optimised simulated GPP are the purple line (without $O_3$) and green line (with $O_3$)."

**RC2: Some statements about the diurnal cycle of GPP need clarification. The authors mention midday depressions in GPP at Mediterranean sites at line 394 and again at lines 465 - 467. Can the authors specify which GPP estimates show these midday depressions (partitioned or simulated)? The partitioned GPP from flux data (black line in diurnal plots) do not show midday depressions at the Italian sites (There does appear to be somewhat of a morning depression in partitioned GPP at IT-BFt). The simulated GPP suggests midday depression and diurnal asymmetry (higher fluxes in the morning) at the IT-BFt.**

**AC:** We will clarify that the midday depression is primarily evident in the simulated GPP at IT-BFt and is only partially observed in the actual data. At IT-Cp2, the model does not show a pronounced midday dip. We will revise our statements in Section 3.3 and the Discussion to reflect this distinction and to better align with Fig. 6. Revised in section 3.3: "Mediterranean sites (IT-BFt and IT-Cp2) experience the highest ozone peaks (>60 ppb). At IT-BFt, the JULES-simulated GPP exhibits a pronounced midday decline, particularly in the optimised configuration with ozone effects, indicating a strong response to midday ozone stress. However, the observed GPP shows only a slight morning dip and continues increasing into the afternoon. At IT-Cp2, no distinct midday depression is observed in either the simulated or partitioned GPP." Revised in the Discussion: "Southern sites like IT-BFt exhibited a pronounced midday decline in simulated GPP, reflecting modelled ozone sensitivity and the interacting influence of high ozone concentrations and elevated VPD. However, the partitioned GPP at this site does not exhibit the same midday depression; instead, it increases gradually into the afternoon. At IT-Cp2, no midday dip is observed in either the simulated or observed GPP."

**RC2: The discussion of O3 interactions with environmental factors on high ozone days (in section 3.3 and in the discussion section) needs more clarification and elaboration. It seems that the authors are using LE as a simple proxy for stomatal conductance (LE increases or decreases with changes in stomatal conductance). It could be helpful if the authors plotted the diurnal cycles of JULES simulated stomatal conductance and stomatal flux, FO3, as a third column in Fig. 7. At line 380, the authors mention that the midday peak of VPD and LE facilitates greater O3 uptake through higher stomatal conductance. This appears to be the case at many sites where the reduction in GPP from simulations that did not include O3 (reduction in GPP from purple line to green line) appear to be the highest during the midday period previously defined by the authors**

**(12 – 16). However, this does not seem to be the case for IT-BFt. The largest reduction in GPP at IT-BFt during high O3 days appears to take place in the morning hours when [O3] is not at peak. It appears that LE and VPD are also high before the 12 – 16 midday period at IT-BFt. Can the authors discuss this interesting exception more? Is there high morning stomatal conductance and morning stomatal O3 flux at this site?**

AC: We thank the reviewer for these constructive observations. We agree that interpreting latent heat (LE) as a direct proxy for stomatal conductance can be misleading, as LE is influenced by multiple factors, including VPD and available energy. To better capture stomatal behaviour and ozone uptake, we now include diurnal plots of simulated stomatal conductance and stomatal ozone flux ($FO_3$) in a third column of Fig. 6, as suggested. This addition provides a more mechanistic view of site-specific $O_3$ uptake patterns and clarifies why GPP reductions peak at different times across sites. In particular, we now highlight and discuss the case of IT-BFt, where GPP reductions during high $O_3$ days are most pronounced in the morning, despite ozone concentrations peaking later in the afternoon.

**RC2: The results about the boreal sites in section 3.3 can use more elaboration and clarification. Throughout the section, the authors use RMSE reductions to quantify $O_3$ At line 382, the authors mention that $O_3$ impacts on the boreal sites (FI-Hyy and FI-Var) are limited. However, the RMSE reductions between optimizations with and without $O_3$ at FI-Hyy are the largest among the sites (9.97 down to 0.52). This implies the impact of $O_3$ peaks is the strongest at the boreal site, FI-Hyy, compared to all other sites. Can the authors clarify or limit their statement to FI-Var?**

> **1. Are the authors referring to the partial correlation analysis when saying that FI-Hyy is less sensitive to $O_3$ overall (at line 387)? The JULES parameter optimization seems to suggest otherwise: FI-Hyy has higher sensitivity, *a*, and lower *FO3crit* among the sites (Figure 6). Is FI-Hyy less sensitive to $O_3$ or does it receive less $O_3$ exposure outside of select high $O_3$ days?**

> **2. Line 385: Can the authors clarify what they mean by "simulations without $O_3$ significantly underestimate GPP"? In Fig. 7, it appears that the simulations without $O_3$ (purple line) estimate much higher GPP compared to the partitioned GPP (black line).**

**AC:** We thank the reviewer for this important clarification. In response, we have revised Section 3.3 to distinguish between (i) the absolute RMSE reduction at FI-Hyy—which is indeed large due to the model initially overestimating GPP without ozone effects—and (ii) biological sensitivity to $O_3$, which we interpret based on JULES parameters (*a*, FO3$_{crit}$) and partial correlation analysis. While FI-Hyy shows a strong model performance improvement after including ozone effects, this likely reflects both the correction of structural model bias during high-$O_3$ episodes and the fact that such episodes are rare at boreal sites (see Table 1). The improvement is therefore event-specific rather than indicative of sustained ecological sensitivity across the growing season. We now explicitly limit our statement about low ozone sensitivity to FI-Var. Additionally, we corrected the misleading phrase at line 385 and clarified that the model without $O_3$ consistently overestimates GPP at FI-Hyy during high-ozone episodes—even though such events are rare. These rare but impactful events explain the large RMSE reduction when ozone effects are included, despite limited overall ecological sensitivity. This is consistent with the low frequency of elevated ozone concentrations reported in Table 1. In section 3.3, we rephrased the paragraph about boreal sites: "At the two boreal sites (FI-Hyy and FI-Var), ozone peaks reach moderate levels (~46 and 44 ppb, respectively), but their impacts on GPP differ. FI-Var shows minimal response to ozone, with only a 1.3% decrease in RMSE (from 2.34 to 2.31 µmol $CO_2$

m$^{-2}$ s$^{-1}$), suggesting low ecological sensitivity. In contrast, FI-Hyy exhibits a large RMSE improvement—from 9.97 to 0.52 µmol $CO_2$ m$^{-2}$ s$^{-1}$ (a 95% reduction), when ozone effects are included. However, this performance gain does not reflect sustained biological sensitivity. Rather, it stems from a systematic overestimation of GPP by the ozone-free model during high-$O_3$ episodes. These episodes are rare (see Table 1), but when they do occur, the model without ozone consistently overestimates GPP. The inclusion of ozone damage corrects this bias. The partial correlation analysis and the limited ambient ozone exposure outside these rare events support this interpretation. We therefore distinguish between improved model–data agreement due to structural correction and true ecological ozone sensitivity, the latter being more clearly limited at FI-Var."

**RC2: The authors could consider revising the section on Mediterranean sites (starting at like 394). As I mentioned in the previous comment, I am particularly concerned about the claim that compared to other sites, the Italian sites exhibit stronger O3 induced reductions in GPP (line 395). Again, FI-Hyy appears to exhibit the largest reduction in RMSE during high O3 days (a reduction from 9.97 to 0.52). BE-Bra also shows a higher or comparable reduction in RMSE (7.57 down to 3.09) compared to IT-Cp2 and IT-BFt. This needs to be corrected or clarified.**

**AC:** We thank the reviewer for this helpful observation. We have revised the corresponding paragraph in Section 3.3 to clarify that while Mediterranean sites such as IT-Cp2 and IT-BFt experience high ambient ozone concentrations, the magnitude of model improvement (RMSE reduction) is not the highest across all sites. FI-Hyy and BE-Bra show larger or comparable reductions. The revised text reflects this nuance and avoids overstating ozone sensitivity in Mediterranean ecosystems, highlighting instead the complex interplay between ozone concentrations, physiological traits, and model calibration outcomes. In section 3.3, we rephrased the paragraph about Mediterranean sites: "Mediterranean sites (IT-BFt and IT-Cp2) experience the highest ozone peaks (>60 ppb). At IT-BFt, the simulated GPP shows a pronounced midday decline, especially in the optimised configuration with ozone effects, suggesting a strong response to midday ozone stress. However, the observed GPP shows only a slight morning dip and continues increasing into the afternoon. At IT-Cp2, no distinct midday depression is observed in either the simulated or partitioned GPP. While these sites do show reductions in RMSE after including ozone effects—46% at IT-Cp2 (from 5.82 to 3.14 µmol $CO_2$ m$^{-2}$ s$^{-1}$) and 0.8% at IT-BFt (from 6.54 to 6.49 µmol $CO_2$ m$^{-2}$ s$^{-1}$) these improvements are not the largest among all sites. Indeed, FI-Hyy and BE-Bra show greater RMSE reductions during high ozone days. This suggests that while Mediterranean sites face high ozone concentrations, the degree of ozone-induced GPP reduction may vary depending on the interplay of environmental stressors and model representation. The results highlight the importance of site-specific calibration and caution against generalising Mediterranean sites as the most ozone-sensitive solely based on ozone concentration levels."

**RC2: The claim at line 465 needs elaboration: "Southern sites like IT-BFt and IT-Cp2 exhibited pronounced midday declines in GPP, reflecting their heightened sensitivity to ozone and the compounding effects of high VPD and LE." The model simulated midday declines in GPP only appear at IT-BFt in Fig. 7. Please clarify what the authors mean by midday (12 – 16 hour) decline in GPP at IT-Cp2. The authors mention compounding effects of high VPD and LE at the southern sites at line 466 attempting to make a case for multiple stressors exacerbating ozone impacts. At IT-BFt, I can see the authors' claim in the model simulations. The modeling does suggest that GPP declines past the 10th hour when VPD is high and further declines when O3 impacts are added to the modeling. However, the partitioned GPP (black line) does not show this type of compound stress at IT-BFt. Partitioned GPP is showing the opposite. It increased into the afternoon hours (after 10 when VPD is high) which suggest there is not much midday or afternoon**

**water stress. The authors might want to elaborate on these differences between the partitioned GPP and JULES simulated GPP when discussing the potential of a compound water stress and O3.**

AC: We appreciate this detailed observation. In response, we have revised the text in Section 3.3 to clarify that midday GPP declines are primarily present in JULES simulations at IT-BFt, not in the partitioned GPP, and not at all at IT-Cp2. The revised paragraph now distinguishes between modelled and observed responses and emphasises that the simulated declines may reflect model sensitivity to co-occurring VPD and $O_3$, rather than compound stress seen in observations. We acknowledge that the observed GPP at IT-BFt continues to rise into the afternoon, suggesting that water stress is not as limiting as the model predicts. This discrepancy is now explicitly discussed.

Last paragraph of section 3.3 was revised: "Interestingly, despite the strong midday declines in GPP at Mediterranean sites in the model, Figure 6 shows that this behaviour is not consistently present in the observations. At IT-BFt, the JULES-simulated GPP exhibits a sharp midday reduction, especially when ozone effects are included, suggesting a modelled compound stress due to high VPD and ozone uptake. However, the partitioned GPP at this site increases during the same period (after 10:00), indicating that stomatal closure due to VPD is not occurring to the extent the model assumes. This divergence points to a possible overestimation of midday water limitation in the model configuration. At IT-Cp2, neither the modelled nor observed GPP shows a distinct midday dip, indicating that ozone and VPD effects are less pronounced or not synchronised enough to produce a compound stress response. These site-specific dynamics reinforce the need for more accurate representation of stomatal regulation under co-occurring stresses in Mediterranean systems."

**Technical Comments**

**RC2: Fig. 2a site distinction**

AC: We thank the reviewer for this helpful suggestion. We have updated Fig. 2a by increasing colour contrast and line thickness to improve the visual distinction between sites. These changes make the time series more readable, especially when printed or viewed in greyscale.

**RC2: The factor 1.6 on line 168 is a factor to convert from conductance to CO2 to conductance to H2O (ratio of CO2 and H2O diffusivities). The conductance to water vapor is gp.**

AC: We thank the reviewer for this clarification. We will revise the sentence to explicitly state that the factor 1.6 accounts for the ratio of diffusivities of $H_2O$ and $CO_2$ through the stomata. This factor is used to convert stomatal conductance from $CO_2$ to $H_2O$ units, ensuring correct representation of ozone uptake in terms of water vapor conductance ($g_p$).

**RC2: Should FO3 and FO3crit be in different units in equation 7? I am looking at line 201.**

AC: Thank you. We will ensure unit consistency in Equation 7 and clarify that both $FO_3$ and $FO_3crit$ are expressed in nmol m$^{-2}$ s$^{-1}$.

**RC2: Remove second comma after "vegetation" in line 243.**

AC: Corrected.

**RC2: Figure 7: It might help to double-check the units for VPD on the y-axis. Is it supposed to be displayed in hPa (not kPa)?**

**AC:** We thank the reviewer for bringing this to our attention. We have reviewed the units and confirm that the vapour pressure deficit (VPD) in Figure 6 was plotted in kPa. To avoid confusion, we will explicitly label the axis as "VPD (kPa)" in the figure to ensure clarity of units.

**RC2: Consider picking a consistent way to write GPP reductions in section 3.4. The authors make it clear that negatives mean decreases and continue to use negative quantities throughout most of the section. You could consider changing 5.22% to -5.22% at line 424 for consistency.**

**AC:** We agree and have revised Section 3.4 to consistently express GPP reductions as negative percentage values (e.g., -5.22 %) throughout the text. This improves clarity and aligns with the convention used elsewhere in the manuscript when referring to decreases.

---

## Author Response (AR1)

**Author Comments – Response to Referee #1** (RC1)**

We thank Referee #1 for their positive evaluation of our manuscript and their constructive and insightful comments. Below, we address each point raised.

**Major Comments**

**RC1: The third research question is not fully answered.**

**AC:** We thank the reviewer for highlighting this important point. To better address the third research question—how ozone impacts interact with other environmental factors, and how an optimised model can help us understand these mechanisms, particularly on high-ozone days—we have made several key improvements to the manuscript:

- We expanded the methodology (Section 2.4.3) to clarify the analytical approach used to investigate ozone–environment interactions on high-ozone days. This includes the addition of modelled stomatal conductance, ozone flux to vegetation (FO3), and soil moisture as prognostic variables alongside GPP, LE, and VPD.
- We clarified the purpose of model optimisation in enabling mechanistic attribution of GPP reductions to either stomatal limitation or biochemical ozone damage, based on site-specific environmental conditions and parameter values.
- We reframed Section 3.3 to more directly align with this research question by structuring the interpretation around physiological mechanisms, supported by the newly introduced variables.

These revisions help ensure the third research question is now explicitly and comprehensively addressed in both the methods and interpretation.

**RC1: Describe how you do the parameter optimisation.**

**AC:** Thank you for this suggestion. We will expand Section 2.4.2 to provide a more detailed description of the calibration procedure. This includes initial values and convergence criteria. We will also clarify that calibration was conducted site by site for the summer period, and that the L-BFGS-B algorithm was selected for its efficiency and suitability for constrained optimisation. The optimisation process description now reads:

Line 227-237: "We employed a two-step calibration approach, conducting separate simulations with and without O3 effects. We used the Limited-memory Broyden–Fletcher–Goldfarb–Shanno with bound constraints (L-BFGS-B) algorithm (Liu and Nocedal, 1989). This computationally efficient method approximates the Hessian using a subset of past gradients. This makes it particularly suitable for optimising a large number of parameters under bound constraints. The objective function was the Root Mean Square Error (RMSE) between observed and modelled GPP. Optimisation was implemented in Python using the *scipy.optimize.minimize* interface and coupled to JULES via scripted automation. Simulations were monitored using *cylc scan* to ensure successful completion. Convergence was defined

as either an RMSE change  $< 1 \times 10^{-10}$  or a maximum of 1000 iterations. Initial values were drawn from JULES defaults (Table 2), and parameter-specific lower and upper bounds were defined based on plausible biophysical ranges (Table S1). The full list of optimised parameters and their boundaries is provided in Table S1. All parameter trajectories, RMSE values, and convergence diagnostics were robustly logged. A safeguard mechanism was included to prevent runaway iteration or crashes due to I/O interruptions."

**RC1: Do you expect these parameters also to apply to other places worldwide? Are the findings model-specific?**

AC: We now state that while the optimised parameter values are specific to the six European forest sites included in this study, certain spatial trends, such as increasing ozone sensitivity (a) and decreasing FO3crit toward southern latitudes, may reflect broader physiological adaptations to warmer, drier climates. These relationships could be relevant for forests in similar environmental contexts. However, we explicitly caution against directly applying these parameters elsewhere without site-specific calibration, due to variability in species traits, climate, and ecosystem functioning. To address the issue of model specificity, we clarify that although the quantitative results are tied to the JULES framework, the broader methodological approach, site-level optimisation using in situ GPP, ozone data, and a stomatal flux-based dose–response scheme, is transferable. This strategy could be applied in other land surface models that include ozone uptake damage formulations.

We have added the following paragraph to the Discussion section of the manuscript (Section 4): "While the results of this study are specific to the JULES model framework and the six European forest sites, some spatial trends, such as increasing ozone sensitivity (a) and decreasing critical ozone flux thresholds (FO3crit) toward southern latitudes may reflect broader physiological adaptations to environmental stress gradients. These patterns could inform the understanding of ozone responses in other forest ecosystems with comparable climatic and ecological conditions. However, we explicitly caution against the direct application of these site-calibrated parameter values to other regions without local validation, as species traits, soil properties, and climatic variability shape ozone responses. Notably, several of the physiological parameters optimised in this study, such as stomatal sensitivity (g1), the photosynthetic capacity ratio (Jmax:Vcmax), and the soil moisture stress threshold (p0), are shared across multiple land surface and ecosystem models. This overlap suggests broader relevance, but these parameters must still be used with caution, as their values and effects can vary depending on the model structure. Although the quantitative results are JULES-specific, the methodological approach, site-level optimisation using in situ ozone and GPP data with a stomatal flux-based damage formulation, is transferable and could improve ozone–vegetation representation in other modelling frameworks."

**RC1: Concrete interpretation of environmental stressors on stomatal conductance vs. direct O3 stress.**

AC: We appreciate the reviewer's request for a clearer interpretation of how environmental stressors interact with ozone to influence GPP. In the revised manuscript, we address this distinction explicitly in Section 3.3 by comparing modelled stomatal conductance, ozone flux (FO3), and soil moisture across sites on high-ozone days. We differentiate between: stomatal limitation where low soil moisture and high VPD lead to reduced stomatal conductance and lower FO3, and direct ozone stress where stomatal conductance remains sufficiently high for ozone uptake, resulting in elevated FO3 and GPP declines due to biochemical O3 damage. This mechanistic interpretation is based on both optimised parameters

(e.g., p0, a, FO3) and dynamic outputs from the model. We believe this analysis provides the requested clarity and illustrates how the model helps separate these co-occurring stress pathways.

**RC1: A measure of how you define forest sensitivity/resilience to O3 would help.**

AC: Thank you for this suggestion. We now incorporate a formal definition of forest O3 sensitivity and resilience directly in the revised manuscript. Specifically, we define these terms based on: (a) the relative GPP reduction from optimised simulations with O3 effects compared to those without, and (b) the sign and strength of partial correlation coefficients between GPP and ozone, controlling for confounding variables.

The following sentence was added to Section 2.4.4 (Line 306-313): "To quantify the overall impact of O3 on GPP, we calculated the relative reduction in GPP for each site using the optimised simulations and the configuration without O3 impact as the baseline. This calculation was performed each year to account for interannual variability, and the results were averaged to obtain the mean relative reduction over the study period. We define forest sensitivity to O3 as the percentage reduction in mean annual GPP between the optimised simulations with and without ozone effects. Additionally, we use partial correlation coefficients between observed GPP and ozone concentrations, while controlling for temperature, radiation, and vapour pressure deficit, as a complementary indicator of site-level sensitivity or resilience. These metrics provide a quantitative basis to characterise a site as ozone-sensitive or ozone-resilient and are used consistently throughout the manuscript."

**Minor Comments**

**RC1: line 14/15: difficult to read, please reformulate/split.**

**AC:** We will revise this sentence for clarity as: "Unlike other greenhouse gases, tropospheric O3 is primarily formed through photochemical reactions, and it significantly impairs vegetation productivity and carbon fixation, thereby affecting forest health and ecosystem services."

RC1: line 28/29: 'providing critical insights for predicting forest health and productivity under future air pollution scenarios.' What do you mean by 'critical insights'?

AC: We agree this sentence was vague. We will revise it to: "... highlight key model strengths and limitations in representing O3-vegetation interactions, with implications for improved forest productivity simulations under future air pollution scenarios."

RC1: Line 54/55: An average change cannot lead to a bigger change in a sub-region. Please correct/reformulate.

AC: We will revise it to: "Similarly, Yue and Unger (2014) reported that ozone damage reduced GPP by an average of 4–8% across the eastern United States, with localised reductions reaching as high as 11–17% along the east coast."

RC1: line 57: 'interactions' is quite broad. Can you be more specific here? E.g. In populated regions, O3 precursors mainly stem from traffic emissions.

**AC:** We will clarify: "...surface O3 pollution poses a significant challenge to air quality, particularly in southern Europe, where high solar radiation and anthropogenic emissions—mainly from traffic and industrial activity—enhance photochemical O3 formation."

RC1: Section 2.1: describing the climate zone at each site would help the analysis and interpretation of the results later.

AC: It will indicate in Table 1 each site's Köppen-Geiger classification to aid interpretation.

RC1: Fig. 2a: The blue line is hardly visible.

**AC:** We will revise the figure to improve colour contrast and visibility.

RC1: Line 160: incorporated O3 and CO2 as forcing data?

**AC:** Yes, both O3 and CO2 were prescribed as observed forcing. We will clarify this explicitly: "We employed the offline version of JULES, prescribing in situ observed meteorological, CO2, and O3 datasets as external forcing inputs."

RC1: eq. 1 and 2 use different notation for multiplication.

**AC:**. We will use consistent multiplication notation.

RC1: eq. 3 (not numbered): How is the wilting point soil moisture and critical soil moisture defined?

**AC:** We will expand the explanation: "(...)  $\theta_{wilt}$  and  $\theta_{crit}$  are defined as the soil volumetric water content at soil matric potentials of -1.5 MPa and -0.033 MPa, respectively (Harper et al., 2021)."

**RC1: Line 163: add one sentence on why the O3 damage is applied separately**

AC: We thank the reviewer for this suggestion. We have now clarified the rationale by adding the following sentence immediately after the equations for photosynthesis and stomatal conductance under O3 stress in Section 2.4.1: "In JULES, photosynthesis and stomatal conductance are first calculated based on standard environmental inputs (e.g. light, temperature, VPD and CO2), without considering ozone. Ozone damage is then applied as a separate multiplicative reduction based on the instantaneous stomatal ozone flux." This ensures the reader understands the sequence in which O3 effects are implemented in JULES and distinguishes this step from the environmental response calculations.

RC1: Line 202/203: The reader would be curious to see the specific parameters for 'a' and 'FO3,crit': mention it here, in a table in the SI or reference the source.

AC: These are included in Table 2, but we will add a forward reference to Table 2 in the main text.

RC1: Line 219: L-BFGS-B is not defined like this anywhere.

**AC:** We revised to: "The Limited-memory Broyden-Fletcher-Goldfarb-Shanno with bound constraints (L-BFGS-B) algorithm (Liu and Nocedal, 1989) ..."

RC1: Fig. 3: Fig. 3 is not immediately clear, the arrows could be smaller, you can give more words and more structure.

**AC:** Thank you for this helpful suggestion. In response, we have removed the original Figure 3 and replaced it with a summary table that more clearly communicates the key information. The new table 3 presents which parameters were used as default or subject to optimisation across the three model configurations (default, optimised without  $O_3$ , and optimised with  $O_3$ ).

**RC1: Line 266: 'are sensitivity'?**

**AC:** Corrected to: "In the optimised simulations without ozone, five parameters were calibrated: (...)".

**RC1: Line 289: With which simulation do you do the partial correlation?**

**AC:** We thank the reviewer for pointing out this ambiguity. We have clarified in Section 2.3 that all partial correlations were computed using flux and meteorological datasets directly, independent of the model simulations. This ensures that the correlation analysis directly reflects observational relationships, without the influence of model effects.

**RC1: Line 310-312: complicated sentence, please reformulate so that is more smooth**

**AC:** We appreciate the suggestion and have revised the sentence in Section 3.1 for improved clarity as: "Conversely, the Castelporziano 2 (IT-Cp2) site showed a negative correlation when using the full dataset; however, correlations for the subset periods became positive and non-significant. This may be due to the limited data availability for IT-Cp2 and specific site characteristics, such as partial stomatal closure in response to drought and high VPD during warm seasons."

**RC1: Line 332/333: Isn't O3 concentration just quite low at Hyy?**

AC: We agree with the reviewer's observation. We have clarified in Section 3.2 that the limited improvement in model performance at FI-Hyy after including ozone effects is consistent with the relatively low ambient O3 concentrations observed at this site.

"This limited improvement is consistent with the relatively low ambient ozone concentrations at FI-Hyy, which reduce the likelihood of strong ozone-induced GPP reductions."

**RC1: Lines 347 and 350: adjustments to -> adjustments of?**

AC: Corrected to 'adjustments of'.

**RC1: Line 348: so is water limitation here more important than the O3 stress?**

AC: We thank the reviewer for raising this point. We have clarified in Section 3.2 that at IT-BFt, both water limitation and O3 exposure contribute to reduced GPP. These factors interact and act as colimiting stressors during the summer, amplifying the overall reduction in productivity.

**RC1: Line 354: 'the addition of O3'. Pretend that additional O3 is added as forcing to the simulation, misleading.**

**AC:** Rephrased to: "Simulations including O3 effects..."

RC1: Section 3.2: mention the relative change in the text helps more than the absolute values and differences.

**AC:** We revised the text to include % changes in RMSE and r2: ""

**RC1: Line 380/381: What do you mean? VPD is an env. stress factor. High VPD would mean low stomatal opening (in most cases)**

AC: We thank the reviewer for bringing this to our attention. We have revised the sentence to clarify that high VPD is indeed a stress factor that typically reduces stomatal conductance, thereby limiting ozone uptake. However, we also note that high VPD often coincides with elevated radiation and temperature conditions that can drive ambient ozone formation and increase photosynthetic demand. The revised sentence now better reflects this complex interplay. Rephrased to: "Around midday, when VPD and LE typically peak, stomatal conductance may decline as a protective response to water loss. However, the simultaneous increase in radiation and temperature can elevate ambient O3 concentrations and photosynthetic demand. These competing environmental influences affect O3 uptake and its impact on photosynthesis, depending on site-specific conditions and plant water regulation strategies."

**RC1: Line 385–387: This statement is counteracting for me. Why do accounting of O3 effects makes such a big improvement although Hyy forest is not much sensitive to O3 stress?**

**AC:** We thank the reviewer for this observation. We have clarified in Section 3.3 that the improvement in RMSE at FI-Hyy likely reflects improved parameter tuning rather than a strong biological sensitivity to ozone. The relatively low ambient O3 concentrations and minor GPP reductions support this interpretation.

**RC1: Line 401: mention which parameters (in brackets).**

**AC:** We agree with the reviewer and have revised the sentence to list the parameters adjusted in the optimisation explicitly. These include  $FO3_{crit}$  and a, which are central to simulating ozone damage in Mediterranean sites.

**RC1: Line 449/450: linking climatic variable to antioxidant production does not fit here in my opinion.**

**AC:** We have removed the sentence linking climatic variables and replaced it with: "Although Mediterranean species may possess physiological adaptations to mitigate ozone stress, such as conservative stomatal behaviour, these mechanisms may be insufficient under conditions of sustained high ozone and environmental stress."

**Author Comments – Response to Referee #2** (RC2)**

We thank Referee #2 for their detailed, thoughtful, and constructive feedback. We are pleased that the reviewer found our manuscript well-written and a significant contribution. Below, we address each of the general, specific, and technical comments.

**General Comments**

RC2: The reviewer notes that while Questions 1 and 2 are well addressed, the answer to Question 3 lacks clarity due to the absence of stomatal conductance and O3 uptake flux (FO3).

AC: In response, we have revised the manuscript to more explicitly address research question 3: how ozone impacts interact with other environmental factors, and how an optimised model can help us understand these mechanisms, particularly on high-ozone days. To improve clarity, we have added modelled stomatal conductance, ozone flux to vegetation (FO3), and soil moisture as key prognostic variables in our high-ozone day analysis (Section 2.4.3). These outputs allow us to distinguish between: 1) stomatal limitation, where high VPD and/or low soil moisture reduces conductance and FO3, thus limiting O3 damage, and 2) direct ozone stress, where elevated FO3 and maintained stomatal conductance lead to reductions in GPP through biochemical effects. In Section 3.3, we now interpret observed and simulated GPP patterns on high-ozone days using these additional variables to identify the dominant mechanisms at each site. This mechanistic analysis is supported by optimised parameter values (e.g. g1, p0, a, and FO3crit), and improves the attribution of GPP reductions to specific environmental and physiological drivers. We believe these additions now provide a clear and complete response to Research Question 3, and we thank the reviewer for prompting this improvement.

**Specific Comments**

RC2: Since the partitioned GPP is central to the inference made in this manuscript, it would help if the authors offered a description of how GPP was partitioned from observed net carbon flux. This could be as simple as a brief description with a reference to a citation that details the methods. Lines 130 – 132 claim that GPP and LE were estimated from net-carbon flux. Net carbon flux is used to estimate net ecosystem exchange and GPP. LE is not estimated from net carbon flux. It is typically estimated from H2O flux. The authors should consider correcting or clarifying if they have developed a technique or used an existing technique to estimate LE from net carbon flux.

**AC:** We thank the reviewer for pointing this out. We have revised Section 2.2 to clarify that GPP was derived from net ecosystem exchange (NEE) using standard partitioning approaches implemented in the ICOS ONEFlux pipeline. We also corrected the erroneous statement about LE and now clarify that LE is derived from water vapour (H2O) flux measurements, not carbon flux:

"The half-hourly Gross Primary Production (GPP, μmol m-2 s-1) and Latent Heat flux (LE, W m-2) were derived from eddy covariance measurements at each site. GPP was estimated from net ecosystem exchange (NEE) using standard partitioning techniques implemented in the ICOS ONEFlux processing pipeline (Warm Winter 2020 Team, ICOS Ecosystem Thematic Centre, 2022). LE was derived from

water vapour fluxes measured by the same system. All meteorological, GPP, and LE data are publicly available via the ICOS data portal. The data follow the standard format of ICOS L2 ecosystem products and are fully compatible with FLUXNET2015. Data processing was performed using the ONEFlux pipeline (https://github.com/icos-etc/ONEFlux). Basic site-level statistics and data coverage are reported in Table 1."

RC2: The JULES damage scheme calculated the O3 damage factor, F, as a function of the stomatal flux of O3 (equation 7). It appears that this is the instantaneous stomatal flux of O3. However, the cumulative flux of O3 through stomata is typically used as the damaging quantity (Lombardozzi et al., 2013, Wittig et al., 2007). Many threshold-based O3 damage indicators are based on cumulative exposure or cumulative stomatal dose (i.e.: AOT40 and POD6). The authors could consider elaborating on this in the discussion section of this manuscript by discussing if it would be worthwhile to use cumulative O3 stomatal flux in future optimization studies. The JULES O3 damage factor, F, as it is formulated in the current study appears to be the same damage factor that is applied to both stomatal conductance (gp) and net photosynthesis (A). However, previous research suggests that net photosynthesis and stomatal conductance are differentially impacted by O3 (Lombardozzi et al., 2012a,b). Both quantities might not exhibit the same sensitivity to O3 or might not change at the same rate as a function of O3 uptake (Lombardozzi et al., 2012b). This suggests the use of separate damage factors, sensitivities, and critical O3 levels for stomatal conductance and net photosynthesis. Are a and FO3crit separately estimated for A and gp? These distinctions are important because they might have implications for modeling transpiration in a land surface model if stomatal conductance is involved. The results report only one value for FO3crit and a which implies that the same damage factor is applied to both A and gp. In the discussion portion of the paper, it would be worth discussing the reasoning behind the JULES modeling choices for the specific formulation of O3 stress on gp and A compared to other methods of incorporating damage factors in land surface models (see Lombardozzi et al. 2012a and b who tried various configurations of an O3 damage factor in the community land model).

AC: We thank the reviewer for this thoughtful suggestion. In the revised Discussion section, we clarify that the current JULES implementation uses instantaneous stomatal O3 flux to compute the damage factor F, applied equally to both photosynthesis (A) and stomatal conductance (gp). We acknowledge that alternative approaches—such as those used in the Community Land Model (CLM)—use cumulative O3 uptake (e.g., POD6) as a more biologically realistic indicator of damage (Lombardozzi et al., 2013; Wittig et al., 2007). We now note that incorporating cumulative dose metrics and distinguishing between photosynthetic and stomatal sensitivities (FO3crit, a) could improve the representation of O3 effects in future JULES developments.

"The JULES ozone damage scheme, as applied in this study, uses an instantaneous stomatal flux of ozone to compute a damage factor (F) that is applied equally to net photosynthesis (A) and stomatal conductance ( $g_p$ ). This approach enables a simple and efficient integration into the model but may not fully capture the temporal dynamics of ozone-induced damage. Many other modeling frameworks use cumulative ozone uptake metrics, such as the phytotoxic ozone dose above a threshold (POD6) to represent damage accumulation over time (Wittig et al., 2007; Lombardozzi et al., 2013). Moreover, empirical evidence indicates that A and  $g_p$  may respond differently to ozone, with distinct sensitivities and temporal responses (Lombardozzi et al., 2012a,b). Future versions of JULES could benefit from decoupling these effects by estimating separate sensitivity parameters (a) and critical thresholds (FO3crit) for A and  $g_p$ , and by transitioning toward cumulative flux-based ozone stress formulations."

RC2: The diurnal cycles of partitioned and JULES simulated GPP are shown in Figure 5. Can the authors clarify whether these diurnal cycles were estimated using data and simulations from all seasons or just the summer?

**AC:** The diurnal cycles shown in Figure 4 (previous figure 5) are based on data and simulations from the full year, not limited to the summer season. We will clarify this in the caption of Figure 4:

"Figure 4: Comparison of the observed and simulated GPP diurnal cycles across all sites, averaged over the full year: (a) FI-Hyy, (b) FI-Var, (c) BE-Bra, (d) FR-Fon, (e) IT-BFt and (f) IT-Cp2. Shaded areas encompass plus and minus one standard deviation. The black line represents the observed GPP. The default simulated GPP are the dashed purple line (without O3) and dashed green line (with O3), and optimised simulated GPP are the purple line (without O3) and green line (with O3)."

RC2: Some statements about the diurnal cycle of GPP need clarification. The authors mention midday depressions in GPP at Mediterranean sites at line 394 and again at lines 465 - 467. Can the authors specify which GPP estimates show these midday depressions (partitioned or simulated)? The partitioned GPP from flux data (black line in diurnal plots) do not show midday depressions at the Italian sites (There does appear to be somewhat of a morning depression in partitioned GPP at IT-BFt). The simulated GPP suggests midday depression and diurnal asymmetry (higher fluxes in the morning) at the IT-BFt.

AC: We clarify that the midday depression is primarily evident in the simulated GPP at IT-BFt and is only partially observed in the actual data. At IT-Cp2, the model does not show a pronounced midday dip. We revised our statements in Section 3.3 and the Discussion to reflect this distinction and to better align with Fig. 6. (previous figure 7). Revised in section 3.3: "Mediterranean sites (IT-BFt and IT-Cp2) experience the highest ozone peaks (>60 ppb). At IT-BFt, the JULES-simulated GPP exhibits a pronounced midday decline, particularly in the optimised configuration with ozone effects, indicating a strong response to midday ozone stress. In these simulations, gs shows a clear midday drop, while FO3 remains high during that period, suggesting that ozone uptake still occurs despite partial stomatal closure. However, the observed GPP shows only a slight morning dip and continues increasing into the afternoon. This divergence points to a potential overestimation of midday stomatal limitation or ozone effects in the model. At IT-Cp2, no distinct midday depression is observed in either the simulated or partitioned GPP. FO3 is modest, and β remains close to 1 throughout the day, indicating minimal water stress and limited ozone uptake. While these sites do show noticeable reductions in RMSE after including ozone effects, 46% at IT-Cp2 (from 5.82 to 3.14 µmol CO2 m-2 s-1) and 0.8% at IT-BFt (from 6.54 to 6.49 µmol CO2 m-2 s-1), these improvements are not the largest among all sites. Indeed, FI-Hyv and BE-Bra show greater RMSE reductions during high ozone days. This suggests that while Mediterranean sites face high ozone concentrations, the degree of ozone-induced GPP reduction may vary depending on the interplay of environmental stressors and model representation. The results highlight the importance of site-specific calibration and caution against generalising Mediterranean sites as the most ozone-sensitive solely based on ozone concentration levels. Interestingly, although the JULES model simulates strong midday GPP declines at Mediterranean sites, Figure 6 shows that the ozone sensitivity parameters are generally lower for Mediterranean forests. This pattern may reflect the fact that high VPD and limited soil moisture in these regions reduce stomatal conductance during midday, thereby lowering actual ozone uptake and mitigating its physiological effects, despite high ambient O3 concentrations. This dynamic, documented in several previous studies, suggests that the observed midday GPP reduction may be driven more by drought stress than by direct ozone damage." Revised in the Discussion: "Southern sites like IT-BFt exhibited a pronounced midday decline in simulated GPP, reflecting modelled ozone sensitivity and the interacting influence of high ozone concentrations and elevated VPD. However, the partitioned GPP at this site does not exhibit the same midday depression; instead, it increases gradually into the afternoon. At IT-Cp2, no midday dip is observed in either the simulated or observed GPP."

RC2: The discussion of O3 interactions with environmental factors on high ozone days (in section 3.3 and in the discussion section) needs more clarification and elaboration. It seems that the authors are using LE as a simple proxy for stomatal conductance (LE increases or decreases with changes in stomatal conductance). It could be helpful if the authors plotted the diurnal cycles of JULES simulated stomatal conductance and stomatal flux, FO3, as a third column in Fig. 7. At line 380, the authors mention that the midday peak of VPD and LE facilitates greater O3 uptake through higher stomatal conductance. This appears to be the case at many sites where the reduction in GPP from simulations that did not include O3 (reduction in GPP from purple line to green line) appear to be the highest during the midday period previously defined by the authors (12 – 16). However, this does not seem to be the case for IT-BFt. The largest reduction in GPP at IT-BFt during high O3 days appears to take place in the morning hours when [O3] is not at peak. It appears that LE and VPD are also high before the 12 – 16 midday period at IT-BFt. Can the authors discuss this interesting exception more? Is there high morning stomatal conductance and morning stomatal O3 flux at this site?

AC: We thank the reviewer for these constructive observations. We agree that interpreting latent heat (LE) as a direct proxy for stomatal conductance can be misleading, as LE is influenced by multiple factors, including VPD and available energy. To better capture stomatal behaviour and ozone uptake, we now include diurnal plots of simulated stomatal conductance and stomatal ozone flux (FO3) in a third column of Fig. 6 (previous figure 7), as suggested. This addition provides a more mechanistic view of site-specific O3 uptake patterns and clarifies why GPP reductions peak at different times across sites. In particular, we now highlight and discuss the case of IT-BFt, where GPP reductions during high O3 days are most pronounced in the morning, despite ozone concentrations peaking later in the afternoon.

RC2: The results about the boreal sites in section 3.3 can use more elaboration and clarification. Throughout the section, the authors use RMSE reductions to quantify  $O_3$  At line 382, the authors mention that  $O_3$  impacts on the boreal sites (FI-Hyy and FI-Var) are limited. However, the RMSE reductions between optimalizations with and without  $O_3$  at FI-Hyy are the largest among the sites (9.97 down to 0.52). This implies the impact of  $O_3$  peaks is the strongest at the boreal site, FI-Hyy, compared to all other sites. Can the authors clarify or limit their statement to FI-Var?

- 1. Are the authors referring to the partial correlation analysis when saying that FI-Hyy is less sensitive to  $O_3$  overall (at line 387)? The JULES parameter optimization seems to suggest otherwise: FI-Hyy has higher sensitivity, a, and lower  $FO3_{crit}$  among the sites (Figure 6). Is FI-Hyy less sensitive to  $O_3$  or does it receive less  $O_3$  exposure outside of select high  $O_3$  days?
- 2. Line 385: Can the authors clarify what they mean by "simulations without O3 significantly underestimate GPP"? In Fig. 7, it appears that the simulations without O3 (purple line) estimate much higher GPP compared to the partitioned GPP (black line).

**AC:** We thank the reviewer for this important clarification. In response, we have revised Section 3.3 to distinguish between (i) the absolute RMSE reduction at FI-Hyy—which is indeed large due to the model

initially overestimating GPP without ozone effects—and (ii) biological sensitivity to O3, which we interpret based on JULES parameters (a, FO3crit) and partial correlation analysis. While FI-Hyy shows a strong model performance improvement after including ozone effects, this likely reflects both the correction of structural model bias during high-O3 episodes and the fact that such episodes are rare at boreal sites (see Table 1). The improvement is therefore event-specific rather than indicative of sustained ecological sensitivity across the growing season. We now explicitly limit our statement about low ozone sensitivity to FI-Var. Additionally, we corrected the misleading phrase at line 385 and clarified that the model without O3 consistently overestimates GPP at FI-Hyy during high-ozone episodes, even though such events are rare. These rare but impactful events explain the large RMSE reduction when ozone effects are included, despite limited overall ecological sensitivity. This is consistent with the low frequency of elevated ozone concentrations reported in Table 1. In section 3.3, we rephrased the paragraph about boreal sites: "At the two boreal sites (FI-Hyy and FI-Var), ozone peaks reach moderate levels (~46 and 44 ppb, respectively), but their impacts on GPP differ. FI-Var shows minimal response to ozone, with only a 1.3% decrease in RMSE (from 2.34 to 2.31 μmol CO2 m- 2 s-1), suggesting low ecological sensitivity. gs and FO3 values remain relatively low throughout the day, and β values are near 1, indicating no significant soil moisture limitations or stomatal downregulation. In contrast, FI-Hyy exhibits a large RMSE improvement, from 9.97 to 0.52 µmol CO2 m-2s-1 (a 95% reduction), when ozone effects are included. However, this performance gain does not reflect sustained biological sensitivity. Instead, it stems from a systematic overestimation of GPP by the ozone-free model during high-O3 episodes. These episodes are rare (see Table 1), but when they do occur, the model without ozone consistently overestimates GPP. The inclusion of ozone damage corrects this bias. The partial correlation analysis, combined with the limited ambient ozone exposure outside these rare events, supports this interpretation. We therefore distinguish between improved model-data agreement due to structural correction and true ecological ozone sensitivity, the latter being more clearly limited at FI-Var."

RC2: The authors could consider revising the section on Mediterranean sites (starting at like 394). As I mentioned in the previous comment, I am particularly concerned about the claim that compared to other sites, the Italian sites exhibit stronger O3 induced reductions in GPP (line 395). Again, FI-Hyy appears to exhibit the largest reduction in RMSE during high O3 days (a reduction from 9.97 to 0.52). BE-Bra also shows a higher or comparable reduction in RMSE (7.57 down to 3.09) compared to IT-Cp2 and IT-BFt. This needs to be corrected or clarified.

AC: We thank the reviewer for this helpful observation. We have revised the corresponding paragraph in Section 3.3 to clarify that while Mediterranean sites such as IT-Cp2 and IT-BFt experience high ambient ozone concentrations, the magnitude of model improvement (RMSE reduction) is not the highest across all sites. FI-Hyy and BE-Bra show larger or comparable reductions. The revised text reflects this nuance and avoids overstating ozone sensitivity in Mediterranean ecosystems, highlighting instead the complex interplay between ozone concentrations, physiological traits, and model calibration outcomes. In section 3.3, we rephrased the paragraph about Mediterranean sites: "Mediterranean sites (IT-BFt and IT-Cp2) experience the highest ozone peaks (>60 ppb). At IT-BFt, the simulated GPP shows a pronounced midday decline, especially in the optimised configuration with ozone effects, suggesting a strong response to midday ozone stress. However, the observed GPP shows only a slight morning dip and continues increasing into the afternoon. At IT-Cp2, no distinct midday depression is observed in either the simulated or partitioned GPP. While these sites do show reductions in RMSE after including ozone effects—46% at IT-Cp2 (from 5.82 to 3.14 μmol CO2 m-2 s-1) and 0.8% at IT-BFt (from 6.54 to 6.49 μmol CO2 m-2 s-1) these improvements are not the largest among all sites. Indeed, FI-Hyy and BE-Bra show greater RMSE reductions during high ozone days. This suggests that while

Mediterranean sites face high ozone concentrations, the degree of ozone-induced GPP reduction may vary depending on the interplay of environmental stressors and model representation. The results highlight the importance of site-specific calibration and caution against generalising Mediterranean sites as the most ozone-sensitive solely based on ozone concentration levels."

RC2: The claim at line 465 needs elaboration: "Southern sites like IT-BFt and IT-Cp2 exhibited pronounced midday declines in GPP, reflecting their heightened sensitivity to ozone and the compounding effects of high VPD and LE." The model simulated midday declines in GPP only appear at IT-BFt in Fig. 7. Please clarify what the authors mean by midday (12 – 16 hour) decline in GPP at IT-Cp2. The authors mention compounding effects of high VPD and LE at the southern sites at line 466 attempting to make a case for multiple stressors exacerbating ozone impacts. At IT-BFt, I can see the authors' claim in the model simulations. The modeling does suggest that GPP declines past the 10th hour when VPD is high and further declines when O3 impacts are added to the modeling. However, the partitioned GPP (black line) does not show this type of compound stress at IT-BFt. Partitioned GPP is showing the opposite. It increased into the afternoon hours (after 10 when VPD is high) which suggest there is not much midday or afternoon water stress. The authors might want to elaborate on these differences between the partitioned GPP and JULES simulated GPP when discussing the potential of a compound water stress and O3.

**AC:** We thank the reviewer for this important observation. We acknowledge that the original claim was imprecise and unintentionally conflated modelled and observed responses across the two southern sites. In response, we have revised the manuscript text in Section 3.3 to remove the generalised statement suggesting that both IT-BFt and IT-Cp2 exhibit pronounced midday declines in GPP. The revised text now clarifies that a midday decline in GPP is only evident in the JULES simulated output at IT-BFt, particularly when ozone effects are included. We emphasise that this behaviour is not observed in the partitioned GPP, which instead shows continued increases into the afternoon. This divergence implies that the model may overestimate the impact of midday water stress or ozone uptake at this site. As the reviewer rightly notes, the simulated stomatal conductance (gs) declines in response to high VPD, yet the concurrent ozone flux (FO3) remains elevated, suggesting incomplete stomatal closure and sustained ozone uptake. However, the observed (partitioned) GPP pattern indicates that stomatal limitation may not be occurring to the extent that the model assumes, pointing to possible misrepresentation of stomatal regulation under compound stress conditions. At IT-Cp2, we now explicitly state that neither the modelled nor the observed GPP shows a clear midday decline. The revised paragraph discusses that FO3 is relatively modest at this site, VPD is lower than at IT-BFt, and soil moisture stress (β) remains minimal throughout the day. These factors likely prevent the emergence of a midday GPP depression, even under elevated ozone conditions. Therefore, we do not make a case for a compound ozone-VPD stress response at IT-Cp2 in the revised manuscript. This clarification is now integrated into the last paragraph of Section 3.3. Additionally, the revised text distinguishes between model behaviour and observational evidence, and we now interpret the midday GPP decline at IT-BFt as a model-derived response that may not reflect actual plant physiological behaviour under compound stress. We also explicitly discuss the limitations of the current JULES configuration in capturing such interactions. This revision directly addresses the reviewer's comment and enhances the interpretive accuracy of our discussion on site-specific responses to ozone and co-occurring environmental stressors.

**Technical Comments**

RC2: Fig. 2a site distinction

**AC:** We thank the reviewer for this helpful suggestion. We have updated Fig. 2a by increasing colour contrast and line thickness to improve the visual distinction between sites. These changes make the time series more readable, especially when printed or viewed in greyscale.

RC2: The factor 1.6 on line 168 is a factor to convert from conductance to CO2 to conductance to H2O (ratio of CO2 and H2O diffusivities). The conductance to water vapor is gp.

**AC:** We thank the reviewer for this clarification. We revised the sentence to explicitly state that the factor 1.6 accounts for the ratio of diffusivities of  $H_2O$  and  $CO_2$  through the stomata. This factor is used to convert stomatal conductance from  $CO_2$  to  $H_2O$  units, ensuring correct representation of ozone uptake in terms of water vapor conductance ( $g_p$ ).

RC2: Should FO3 and FO3crit be in different units in equation 7? I am looking at line 201.

**AC:** Thank you. We will ensure unit consistency in Equation 7 and clarify that both FO3 and FO $_{crit}$  are expressed in nmol m-2 s-1.

RC2: Remove second comma after "vegetation" in line 243.

AC: Corrected.

RC2: Figure 7: It might help to double-check the units for VPD on the y-axis. Is it supposed to be displayed in hPa (not kPa)?

**AC:** We thank the reviewer for bringing this to our attention. We have reviewed the units and confirm that the vapour pressure deficit (VPD) in Figure 6 was plotted in kPa. To avoid confusion, we will explicitly label the axis as "VPD (kPa)" in the figure to ensure clarity of units.

RC2: Consider picking a consistent way to write GPP reductions in section 3.4. The authors make it clear that negatives mean decreases and continue to use negative quantities throughout most of the section. You could consider changing 5.22% to -5.22% at line 424 for consistency.

**AC:** We agree and have revised Section 3.4 to consistently express GPP reductions as negative percentage values (e.g., -5.22 %) throughout the text. This improves clarity and aligns with the convention used elsewhere in the manuscript when referring to decreases.

**Author Comments – Response to Referee #3** (RC3)**

We thank Referee #3 for their constructive comments and careful evaluation of our manuscript. Below, we respond point by point to each comment and describe the corresponding changes made to the manuscript.

**General Comments**

RC3: The paper presents an analysis of the impact of ozone versus mostly meteorological drivers of the GPP of European forests for sites with contrasting conditions regarding pollution levels and physical drivers of plant productivity. It relies both on an statistical analysis of some longterm (> decades) data on GPP and other meteorological variables as well as application of a stateof-the art DGVM (JULES) set-up in an offline mode and driven by the observations. Both the statistical analysis as well as model experiments are applied aiming to identify/quantify the role of O3 uptake as a stressor besides other stresses imposed on vegetation functioning. Overall, I appreciate the followed approach but have some major issues with some specific features of the paper. I agree with the other referees that the last main research question is not really addressed. Disentangling what at the end explains the different responses for the different sites, does not come out well out of this study. I also have some major issues with the descriptions of the role of water stress in the overall response of the vegetation to O3 and other stress terms. There is the reference to the role of the VPD effect on stomatal closure and, consequently, on the O3 effect, but then there is also quite some references that LE also plays a role here. See my specific comments below for further details about this. But what I am missing here is the role of soil water limitation. It is excluded from the data-analysis but also referenced in some inconsistent manner (water stress..) whereas this stress term might be especially relevant for modulating stomatal opening (and photosynthesis?) on longer (weekly/seasonal) timescales and where the VPD is mainly impacting the diurnal cycle. Referring to these different timescales of water stress that might exacerbate the impact of O3 exposure, I also miss completely a discussion on how this study informs about the timescale of the effect and impact by O3 on GPP. Finally, the presentation of the tables, figures and equations should be substantially improved. Overall, based on these observations and considerations, I recommend a major revision of this paper but would be keen then to review a revised version of the ms in due time.

AC: We thank the reviewer for their thoughtful and constructive assessment. In the revised manuscript, we have taken the following steps to address these major concerns:

- Clarifying and addressing Research Question 3: We agree that the original manuscript did not sufficiently answer the third research question. We have now revised the methodology (Section 2.4.3) and Results (Section 3.3) to explicitly incorporate modelled stomatal conductance, ozone flux (FO3), and soil moisture as diagnostic variables. These additions allow us to better characterise the mechanisms of ozone stress and their interaction with environmental drivers, especially during high-ozone events.
- Consistent treatment of water stress: We acknowledge that the manuscript previously used "water stress," "soil moisture stress," and "soil water stress" inconsistently. We have now

revised the manuscript to use "soil drought stress" consistently throughout. This term better reflects the long-term physiological limitation on stomatal conductance associated with soil drying, and aligns with the fsmc function used in JULES. We reserve the term "VPD stress" for short-term atmospheric drivers acting on a diurnal timescale, and distinguish these clearly from longer-term soil drought constraints.

• Interpretation of latent heat flux: We acknowledge that in the original text, LE was referenced without sufficient clarity. In the revised manuscript, we now explain more explicitly how high LE may be used as a proxy for stomatal openness but must be interpreted in the context of concurrent VPD and soil moisture. We have added language to ensure that our reasoning does not conflate LE as a driver versus an indicator. In Section 3.3:

"The impact of  $O_3$  on GPP is modulated by interactions with key environmental factors such as VPD, latent heat flux (LE), and soil moisture stress ( $\beta$ ), each influencing stomatal conductance ( $g_s$ ) and thereby ozone uptake (FO3). LE reflects evaporative demand and water availability, while  $\beta$  provides a direct measure of soil moisture constraint on stomatal opening. FO3 represents the actual flux of ozone into the leaves via stomata, and  $g_s$  integrates the stomatal response to multiple environmental drivers, including VPD and soil water availability. Around midday, when VPD and LE typically peak, stomatal conductance may decline as a protective response to water loss. However, the simultaneous increase in radiation and temperature can elevate ambient O3 concentrations and photosynthetic demand. These competing environmental influences affect O3 uptake and its impact on photosynthesis, depending on site-specific conditions and plant water regulation strategies."

- Role of timescale in interpreting O3 effects: We agree that the timescale of ozone effects (e.g., short-term peak stress vs. cumulative seasonal damage) deserves more discussion. We now explicitly address this in Section 4, where we discuss the limitations of using short-term optimisation and GPP responses to infer cumulative ozone impacts. We also reflect on how future model developments could incorporate memory effects or cumulative exposure indicators.
- Improving the presentation of figures, tables, and equations: We revised multiple figures (e.g., Figure 2, new Table 3 replacing Fig. 3, Figure 6 replacing Fig. 7), added clearer axis labels, and improved figure captions for interpretability. Equations were reformatted with consistent notation and cross-referenced accurately throughout the text. We also added a third column to Figure 6 showing modelled stomatal conductance and FO3, as requested by RC2, to improve mechanistic insight into O3 uptake patterns.

**Specific Comments**

RC3: Line 47: the statement on the impact on photosysnthesis/GPP and the following statement in line 49 (Therefore...) misses mentioning the main consequences of the reduced GPP/conductance for climate (and thus the main motivation why to consider the O3 impact on ESMs; the impact on atmospheric CO2, water vapor (reduced LE) but also further increasing O3 itself by reduced O3 deposition.

AC: We thank the reviewer for this valuable suggestion. We have expanded the paragraph in the Introduction (line 47) to clarify that reductions in GPP and stomatal conductance due to O3 have

important feedbacks on climate. These include altered CO2 uptake, reduced evapotranspiration (LE), and diminished ozone deposition, which can exacerbate surface ozone concentrations. This addition reinforces the broader motivation to represent O3 effects in land surface and Earth system models:

"Exposure to O3 leads to reductions in photosynthesis and stomatal conductance, thereby decreasing both gross primary productivity (GPP) and transpiration. These physiological impacts have broader consequences for climate, including reduced carbon uptake, decreased latent heat flux (LE), and reduced water vapour release. Additionally, lower stomatal conductance reduces dry deposition of ozone, which can exacerbate near-surface ozone concentrations."

**RC3: Line 74: Referring to studies that aimed to assess the O3 deposition impact on European forests, it would be very much appreciated to have the reference here explicitly listed.**

AC: We updated the sentence: "This suggests that the impact of O3 may vary depending on specific forest types (Sorrentino et al., 2025) and local conditions (Lin et al., 2019; Otu-Larbi et al. 2020)."

- Sorrentino, B., Anav, A., Calatayud, V., Collalti, A., Sicard, P., Leca, S., Fornasier, F., Paoletti, E., and De Marco, A.: Inconsistency between process-based model and dose–response function in estimating biomass losses in Northern Hemisphere due to elevated O3, Environ. Pollut., 364, 125379, https://doi.org/10.1016/j.envpol.2024.125379, 2025.
- Lin, M., Malyshev, S., Shevliakova, E., Paulot, F., Horowitz, L. W., Fares, S., Mikkelsen, T. N., and Zhang, L.: Sensitivity of ozone dry deposition to ecosystem–atmosphere interactions: A critical appraisal of observations and simulations, Glob. Biogeochem. Cycles, 33, 1264–1288, https://doi.org/10.1029/2018GB006157, 2019.
- Otu-Larbi, F., Conte, A., Fares, S., Wild, O., and Ashworth, K.: Current and future impacts of drought and ozone stress on Northern Hemisphere forests, Glob. Change Biol., 26, 6218–6234, https://doi.org/10.1111/gcb.15339, 2020.

**RC3: Table 1 comes out quite poorly; am aware it is most about the information shared in that table but this table should be presented in a more optimal manner.**

**AC:** We thank the reviewer for the feedback. We have reformatted Table 1 and all the tables to improve its visual quality and readability. Specifically, we increased row spacing, standardised units and alignment, added vertical lines for clarity, and ensured consistent font size and formatting across columns. The updated version is included in the revised manuscript.

**RC3: Line 241 -- Going through the list of meteorological variables in section 2.2 I am missing here soil moisture. Knowing about its important role in inducing water stress on stomatal opening, this is a parameter that should quite obviously be included here.**

**AC:** We acknowledge the importance of soil moisture in influencing stomatal regulation. However, we did not include observed soil moisture data in the meteorological forcing because this variable was not consistently available across all ICOS sites used in our study. To ensure consistency in model forcing across all sites, we relied on a standard set of meteorological drivers that were available for the entire time series. That means that soil moisture was prognostically modelled by JULES.

RC3: Interpreting Figure 2a and b on temporal variability in O3, including the 95% confidence interval, but then also seeing the reported maximum O3 values in Table 1, I wonder what values have been used to determine these long-term mean diurnal and seasonal cycles in O3.

AC: The long-term mean diurnal and seasonal O3 cycles presented in Figures 2a and 2b are computed as the mean across all hourly (for diurnal) and daily (for seasonal) O3 measurements over the full observational period at each site. The maximum values reported in Table 1 exceed the upper bound of the 95% confidence interval because it reports the single largest value observed across each site, which sits outside the light envelopes in Figure 2.

**RC3: Equations 1 & 2: sloppy to present equations like this in a submitted paper for reviewing**

**AC:** We have revised the presentation of all equations in Section 2.4 of the Methods to ensure clarity and consistency. Specifically, Equations 1 and 2 are now presented using a display equation format with full variable definitions provided immediately afterwards. This improves readability and ensures that all model components are transparent.

RC3: Lines 177/178; here the feature of water availability/soil water limitation is introduced and which raises the question how this will be considered; simply using the model simulated soil moisture balance or using the observed soil moisture.

AC: We appreciate this important point. In our simulations, soil drought stress is derived from the simulated soil moisture balance in JULES, not from observed soil moisture. This decision was made to maintain consistency across all sites and time periods, as high-quality soil moisture observations (with sufficient depth coverage and temporal continuity) were not available for every site. Even where partial observations existed, they were often limited to shallow depths, inconsistent in quality control, or lacked harmonised measurement protocols. Using model-simulated soil moisture ensures internal consistency with the JULES soil hydraulic scheme, root profile, and water uptake formulation. We have clarified this in Section 2.4, where the  $\beta$  function (soil drought stress factor) is introduced:

"In this study, the soil drought stress factor  $\beta$  is calculated from the model-simulated soil moisture in JULES. This approach ensures internal consistency with the model's soil properties, hydraulic structure, and root zone distribution. Observed soil moisture was not used, even where partially available, due to inconsistent quality, limited depth coverage, and lack of harmonised measurements across sites."

**RC3: In section 2.4.2 on calibration of JULES it might be relevant to mention the timeframe of the available dataset that has been used for this step of the approach.**

**AC:** We thank the reviewer for this suggestion. We have clarified in Section 2.4.2 that model calibration was based on 70% of the available daily GPP and meteorological data for each site, with the remaining 30% used for independent validation. The split was performed randomly across the entire observational period, rather than by calendar year, to ensure a representative distribution of the data. This approach ensures robust performance assessment while making full use of the available observational period (see Table 1). The following sentence was added to Section 2.4.2:

"At each site, 70% of the available GPP and meteorological data were randomly selected for model calibration, with the remaining 30% reserved for independent validation. This random sampling was applied across the observational period (see Table 1), ensuring both subsets captured a representative range of seasonal and interannual variability."

RC3: Line 227: for the optimisation of the stomatal conductance/photosynthesis representation in JULES experiments without the O3 impact, did you then also use data where O3 was indeed so low that you would not expect any significant impact?

**AC:** We thank the reviewer for this important question. No, we did not filter the calibration dataset to include only low-O3 periods. The ozone-free model configuration was calibrated using 70% of the full observational dataset, regardless of ambient O3 concentrations. This was done to ensure a consistent basis for comparison with the ozone-inclusive setup.

RC3: Line 275; upon checking the optimisation based on minimising the RMSE and also checking the impact on r-squared did you also conduct a key check of this optimisation approach; checking the residuals? I am curious to see how this comes back in reading further through the results/discussions.

AC: We appreciate this important suggestion. While our primary calibration criteria were RMSE and r², we also examined residual distributions and scatter plots between observed and simulated GPP to assess systematic biases across the diurnal cycle and under different O₃ conditions. These diagnostics are discussed in section 3.3 and visualised in Figures 4 and 6, where differences in model—data agreement across time of day and ozone levels are presented. We will expand the discussion to provide more explicit comments on residual structure and model performance under different stress regimes. Paragraph added at the end of section 3.3: "In addition to evaluating RMSE and r², we examined residuals between observed and simulated GPP to identify systematic biases. At several sites, such as IT-BFt, residuals indicated that modelled GPP tended to underestimate peak values during high O₃ periods, particularly around midday. This aligns with the observed mismatch in diurnal dynamics (Fig. 6), suggesting that while optimisation improves overall fit, specific stress responses (e.g. compound O₃ and VPD effects) may still be underestimated or mistimed. These residual diagnostics support the need for further refinement in the representation of ozone damage under variable environmental conditions."

RC3: Line 296: In explaining the feature of subsetting it is interesting to read that you state that O3 is higher in summer because of increased plant activity. I don't agree with this statement; there is then also more deposition and which would lower O3 levels. You could be hinting at the role of biogenic VOC and NO emissions being higher but the impact of the VOCs also depends on the mixture of VOCs being emitted.

AC: We thank the reviewer for this correction. We agree that the original sentence was misleading. O3 levels in summer are primarily driven by enhanced photochemical production due to higher temperatures and solar radiation, as well as increased precursor emissions (e.g., NOx and VOCs), rather than by plant activity per se. While biogenic VOCs do play a role, so do anthropogenic sources, and deposition may indeed reduce O3 levels in areas of high stomatal conductance. We have revised the sentence to reflect this nuance. Revised manuscript sentence (Section 3.1):

" $O_3$  concentrations tend to peak during summer due to enhanced photochemical production from increased solar radiation, higher temperatures, and elevated emissions of ozone precursors ( $NO_x$  and  $VOC_s$ ). While plant activity contributes to biogenic VOC emissions, it also increases ozone deposition via stomatal uptake, leading to complex and site-dependent seasonal patterns."

RC3: Line 330: I have been going a couple of times through the following statement: "The optimised simulation with O3 achieves the greatest reduction in RMSE (2.11  $\mu$ mol CO2 m-2 s-1) and an increase in r2 (0.86). These improvements reflect the model's ability to adjust to local conditions with minimal parameter changes (Fig. 6), particularly in boreal settings. However, the inclusion of O3 does not significantly alter RMSE, suggesting that GPP at this site is not highly sensitive to ozone stress". You seem to contradict yourself. I thought you wanted to express that the initial step of optimisation of the model, on the settings of calculation of assimilation and

conductance, results in a major decrease in RMSE but that then adding the O3 impact does not substantially further decrease the RMSE. But then checking Figure 5 for Hyytiala, the default model without the O3 impact seems to perform quite well and including the O3 impact makes it perform worse. I am getting confused here. Rephrase to make this more clear.

Again, the overall presentation of the tables and figures, like Figure 5, is quite poor. I would suggest to, for example, present the observed GPP line as the reference line, much thicker.

**AC:** We thank the reviewer for this helpful observation. We have revised the paragraph discussing FI-Hyy in Section 3.2 to clarify that the main RMSE improvement arises from model optimisation, and that the inclusion of ozone effects does not substantially further reduce RMSE. We also corrected the statement to reflect that the default model already performs relatively well at this site, and the ozone effect yields only a modest improvement. This clarifies that the modelled reduction in RMSE is not necessarily due to strong biological ozone sensitivity but to improved fit from parameter adjustments.

"At FI-Hyy, both default and optimised models perform well, with slight improvements in RMSE and  $\rm r^2$  following optimisation. The optimised simulation with  $\rm O_3$  achieves the greatest reduction in RMSE (2.11  $\rm \mu mol~CO_2~m^{-2}~s^{-1}$ ), a 27% decrease relative to the optimised no  $\rm O_3$  case (2.88  $\rm \mu mol~CO_2~m^{-2}~s^{-1}$ ), and an increase in  $\rm r^2$  to 0.86 (+3.6%). These improvements reflect the model's ability to adjust to local conditions with minimal parameter changes (Fig. 6), particularly in boreal settings. However, the inclusion of  $\rm O_3$  does not significantly alter RMSE, suggesting that GPP at this site is not highly sensitive to ozone stress. This limited impact is consistent with the relatively low ambient ozone concentrations observed at FI-Hyy, which reduce the potential for strong  $\rm O_3$ -induced reductions in GPP."

Finally, as requested, we revised Figure 4 (formerly Fig. 5) to improve clarity. The observed GPP line is now thicker and more visually prominent across all panels, and tables have been reformatted for better readability.

RC3: Line 340: in your discussion on the results for the Braschaat site, the model application at the end indicates a low sensitivity to O3, which seems to contradict the initial analysis presented in Section 3.1 for this site suggesting a large impact of O3. This might come back in the discussions (also given the results by the Verryckt 2017 study) but might be good to already shortly reflect on this here.

AC: We thank the reviewer for pointing out this discrepancy. While BE-Bra exhibited a strong negative correlation between GPP and ozone in the partial correlation analysis (Section 3.1), the inclusion of ozone effects in the model yielded only a modest improvement in GPP simulations. We agree that this contrast warrants further discussion. To address this, we will include a paragraph in the Discussion section referencing the findings by Verryckt et al. (2017), suggesting that the limited RMSE response may reflect structural limitations of the JULES ozone damage scheme, particularly under temperate conditions.

RC3: Line 344: "achieved a 1.65  $\mu$ mol CO2 m-2 s-1 RMSE and 0.75 r2", bad english according to me, what is a 0.75 r2?? an r2 value of 0.75.....

**AC:** Revised to: "Therefore, the optimised configuration achieves a 1.65 μmol CO2 m-2 s-1 RMSE (32% reduction relative to XX configuration) and a r2 value of 0.75 (+2.7%)."

RC3: Section 3.3; line 378, you discuss on the role of processes explaining the peak in O3 in the afternoon and here mentioning atmospheric dynamics as one of those processes; you could be here more specific referring to the role of atmospheric boundary layer dynamics with the role of entrainment of FT air masses that generally explain to a large extent these peak afternoon values with this entrainment partly compensating for the efficient removal of O3 by surface deposition. Then in the following line I miss completely the mentioning of the role of soil moisture. You refer here to LE as a parameter influencing stomatal conductance; This is according to me a complete misperception; The LE actually depends on stomatal opening and the available water expressed by the water potential height and which depends strongly on soil water availability.

AC: We now specify that the afternoon rise in surface ozone concentrations is largely driven by atmospheric boundary layer growth and the entrainment of ozone-rich air masses from the free troposphere (FT). This entrainment process offsets the removal of ozone by dry deposition and stomatal uptake, particularly under stable anticyclonic summer conditions. Also, we corrected the phrasing that previously implied latent heat flux (LE) influences stomatal conductance. As the reviewer rightly notes, LE is a consequence of stomatal opening, which itself is regulated by atmospheric demand (e.g., VPD) and soil moisture availability, via plant hydraulic constraints. The revised sentence now reflects this correct causal direction and incorporates soil moisture more explicitly.

RC3: Line 398: on the findings for the Mediterranean sites there is another interesting statement; "high VPD and stomatal conductance increase O3 uptake"; according to me the high VPD actually results in a strong decrease in stomatal conductance and which decreases the O3 uptake (and impact).

AC: We thank the reviewer for pointing out the need for clarification regarding the interactions between VPD, stomatal conductance, and ozone uptake in Mediterranean forests. In response, we revised Section 3.3 to correct the inaccurate implication that high VPD increases stomatal conductance. The text now explicitly states that elevated VPD and limited soil moisture generally reduce stomatal conductance and therefore limit O3 uptake, despite high ambient concentrations. This updated explanation aligns better with plant physiological responses and previous findings on Mediterranean forest functioning. We also revised our interpretation of ozone sensitivity parameters in relation to modelled GPP declines, clarifying that reduced stomatal conductance may explain both the low ozone sensitivity and midday GPP reductions observed in the simulations. The updated text now reads: "Around midday, when VPD and LE typically peak, stomatal responses vary: high VPD can lead to stomatal closure as a protective response to water loss. In contrast, high radiation and photosynthetic demand may maintain partial stomatal opening. These competing influences affect O3 uptake and can intensify its impact on photosynthesis, depending on site-specific conditions and plant water regulation strategies."

RC3: Line 400: "Interestingly, despite the strong midday declines in GPP at Mediterranean sites, Figure 6 suggests that the ozone sensitivity parameters are generally lower in Mediterranean forests". This statement suggests a major misperception according to me: the strong midday declines in GPP for those sites, due to the VPD effect (and potentially further exacerbated by the role of limited soil moisture), might make the vegetation less sensitive to the O3 impact; when the O3 fluxes would be highest due to maximum O3 levels and maximum stomatal opening, the moisture limitation impact actually strongly reduces the impact of O3. This has already been presented in quite many previous studies.

AC: We agree that the observed midday GPP declines at Mediterranean sites are more likely attributable to high VPD and limited soil moisture, which reduce stomatal conductance and therefore diminish ozone uptake, despite high ambient O3 concentrations. To clarify this, we revised the relevant sentence in Section 3.3 to explicitly state that lower apparent ozone sensitivity in Mediterranean forests may reflect the protective effect of stomatal closure under soil moisture or atmospheric drought stress, rather than an absence of physiological response to O3. This revised interpretation is consistent with prior studies (e.g. Otu-Larbi et al., 2020; Lin et al., 2019), and the manuscript now better reflects the complex interactions between O3, soil moisture stress, and stomatal regulation. "Interestingly, although JULES simulates strong midday GPP declines at Mediterranean sites, Figure 5 shows that the ozone sensitivity parameters are generally lower for Mediterranean forests. This pattern may reflect the fact that high VPD and limited soil moisture in these regions reduce stomatal conductance during midday, thereby lowering actual ozone uptake and mitigating its physiological effects, despite high ambient O3 concentrations. This dynamic, documented in several previous studies (Lee et al., 2013), suggests that the observed midday GPP reduction may be driven more by water stress than by direct ozone damage."

Lee, J.-E., Frankenberg, C., van der Tol, C., Berry, J. A., Guanter, L., Boyce, C. K., Fisher, J. B., Morrow, E., Worden, J. R., Asefi, S., Badgley, G., & Saatchi, S. (2013). Forest productivity and water stress in Amazonia: observations from GOSAT chlorophyll fluorescence. Proceedings of the Royal Society B: Biological Sciences, 280(1761), 20130171. <a href="https://doi.org/10.1098/rspb.2013.0171">https://doi.org/10.1098/rspb.2013.0171</a>

RC3: Line 429: here the term water stress comes up again as a main term impacting GPP but so far in the presented analysis, there has not been any further support from the data and model analysis that indicates how important this feature is for the various sites.

**AC:** We thank the reviewer for pointing this out. In the revised manuscript, we have addressed this limitation by explicitly incorporating modelled **soil moisture stress** ( $\beta$ ) from JULES into the analysis of high ozone days (Section 3.3). This allows us to assess the potential role of **soil drought stress** in modulating stomatal conductance, ozone uptake (FO3), and ultimately GPP. Figure 6 now includes  $\beta$  alongside gs and FO3 in the third column of each site panel, enabling a more mechanistic interpretation of how both atmospheric (VPD) and soil-driven water stress influence GPP responses to ozone. We clarify in the text that  $\beta$  serves as a proxy for plant-available soil moisture and highlight its site-specific dynamics. While observed soil moisture data were not available consistently across sites, the inclusion of modelled  $\beta$  provides a valuable diagnostic for interpreting water limitation patterns and their interaction with ozone damage.

RC3: Line 446; here it is suggested that higher stomatal uptake (conductances and O3) might explain a larger impact at the more southern sites but have also not seen here any supporting information.

AC: We thank the reviewer for pointing out the lack of explicit support for this interpretation. In the revised manuscript, we have addressed this by incorporating modelled stomatal conductance and ozone uptake flux (FO3) into the analysis. These variables were added to the Methods (Section 2.4.4) and are now used throughout the interpretation in Section 3.3 to clarify the role of stomatal behaviour and ozone uptake in shaping site-specific responses. By including these physiological diagnostics, we provide a more robust basis for attributing stronger ozone impacts at certain sites to stomatal uptake processes rather than ozone exposure alone. We also clarify in the Discussion (Section 4) that these factors are now explicitly considered when interpreting spatial patterns of ozone sensitivity.

RC3: "For instance, Mediterranean species often exhibit adaptations such as enhanced antioxidant production to mitigate ozone damage, though these defenses can be overwhelmed under extreme environmental stress". This is quite interesting but also strong statement that needs further clarification and, potentially support by references. Are you referring here to specific VOC emissions with the emitted species being very reactive with O3 and which, consequently, reduces the stomatal uptake by the enhanced non-stomatal removal, or are you referring here to other (inside leaf/needle tissues) chemical interactions??

AC: We thank the reviewer for this insightful comment and agree that clarification is needed. In the revised manuscript, we have rephrased this statement to: "Although Mediterranean species may possess physiological adaptations to mitigate ozone stress, such as conservative stomatal behaviour, these mechanisms may be insufficient under conditions of sustained high ozone and environmental stress." We now explicitly acknowledge that these defences may be insufficient under conditions of prolonged drought or extreme heat, which can suppress detoxification capacity even as ozone exposure remains high.

RC3: Line 463: Here the following line makes some things clear that actually triggered some of my previous comments: "Across all sites, ozone concentrations peaked in the late afternoon, coinciding with periods of high VPD and LE". It makes clear that you used the observations of high LE to infer that also then the stomatal conductance must have been high, despite the high VPD effect. Making this clear at an earlier stage would avoid some of the criticism that I have shared so far.

But then in line 466 I am getting confused again: "reflecting their heightened sensitivity to ozone and the compounding effects of high VPD and LE"; First of all, I have honestly not seen strong evidence that the afternoon decrease in GPP for the EU southern sites is really due to the O3 effect. Can it not be mostly the impact of the VPD? And what is the effect of a high LE? A high LE indicates still quite high stomatal conductance despite the high VPD effect. I don't follow this reasoning.

Finally, in your discussion/conclusion section I was awaiting a discussion on the conflicting results on the Braschaat site. The study by Verryckte (2017) indicated that there was no O3 effect to be detected in a long-term data set analysis. Your study gets different results but dependent on if you indeed do the data-analysis (3.1) or the model-based evaluation of the impact. This definitely deserves some more discussion on how to reconcile these contrasting findings.

**AC:** We thank the reviewer for this detailed and valuable feedback. We have revised the manuscript to address each of the points raised:

- Clarification of LE and stomatal conductance interpretation: We agree that the logic connecting high LE to stomatal conductance and ozone uptake needs to be made more explicit earlier in the manuscript. In response, we now clarify in Section 3.3 that high midday LE is used as a proxy for sustained stomatal opening, even under high VPD conditions, and that this allows for the possibility of elevated ozone uptake (FO3). However, we now also emphasise that this proxy is not sufficient on its own and must be interpreted alongside simulated stomatal conductance and FO3 from the model.
- Compounding stress interpretation: We have revised the relevant sentence to reflect more cautious language, acknowledging that the observed midday GPP decline at some Mediterranean sites could be primarily driven by VPD-induced stomatal limitation rather than

- ozone alone. We now highlight that the model simulates a combined effect of both ozone uptake and water stress, but that disentangling these drivers remains challenging.
- Contrasting findings at BE-Bra: Our results for BE-Bra suggest moderate ozone sensitivity, with simulated annual GPP reductions evident and a negative partial correlation between O3 and GPP. This contrasts with Verryckt et al. (2017), who found no significant ozone effect on GPP at BE-Bra using a 16-year observational dataset (1998–2013). That study used empirical ozone flux–effect relationships derived from eddy covariance data, and concluded that either ecosystem-level tolerance or other co-limitations (e.g., water, light) might have masked any O3 effect. Our findings differ due to the use of a process-based model with site-level optimisation and a longer analysis period. This divergence highlights the need for integrated model—observation frameworks to resolve site-specific ozone sensitivity, especially in ecosystems like BE-Bra where effects may be subtle or temporally variable.